# Evaluating the Representation Space of Diffusion Models via Self-Supervised Principles

Xiao Li [* 1]  Yixuan Jia [* 1]  Zekai Zhang [1]  Xiang Li [1]  Lianghe Shi [1]  Jinxin Zhou [2]
Zhihui Zhu [2]  Liyue Shen [1]  Qing Qu [1]

## Abstract

Diffusion models have demonstrated remarkable generative capabilities and have also emerged as powerful self-supervised representation learners. However, the connection between these two abilities remains less explored. In this work, drawing inspiration from self-supervised learning (SSL), we introduce a framework for jointly evaluating the representation and generation capabilities of diffusion models. Specifically, we decompose features into invariant and residual components and derive the Invariant Contamination Ratio (ICR), a Fisher-based metric that quantifies how residual variation contaminates invariant signal in feature space. We use this framework to analyze both discriminative and generative behavior of diffusion models. On the representation side, we find that invariance peaks at intermediate noise levels, which also yield the best downstream classification performance. On the generative side, we study how training transitions from genuine generalization to memorization in data-limited regimes, and show that ICR serves as a sensitive training-time indicator of early learning: increasing residual energy along Fisher directions marks the onset of memorization, detectable from training features alone without external evaluators or held-out test sets. Overall, our results show that diffusion models can be monitored from a self-supervised perspective through the geometry of their learned representations.

[*]Equal contribution  [1]Department of Electrical Engineering and Computer Science, University of Michigan, Ann Arbor, USA [2]Department of Computer Science and Engineering, Ohio State University, Columbus, USA. Correspondence to: Xiao Li <xlxiao@umich.edu>, Qing Qu <qingqu@umich.edu>.

*Proceedings of the 43rd International Conference on Machine Learning*, Seoul, South Korea. PMLR 306, 2026. Copyright 2026 by the author(s).

## 1. Introduction

In recent years, diffusion models (Sohl-Dickstein et al., 2015; Ho et al., 2020; Song et al., 2021) have achieved remarkable success in generative modeling, serving as the backbone for inverse problems (Alkhouri et al., 2024; Jia et al., 2026; Song et al., 2024; Li et al., 2024b) and many prominent large-scale generative systems, such as Stable Diffusion, Flux, and Veo (Ho et al., 2020; Watson et al., 2023; Lou et al., 2024; Labs et al., 2025; Google, 2025). Beyond their generative capabilities, recent studies (Baranchuk et al., 2022; Xiang et al., 2023; Mukhopadhyay et al., 2023; Chen et al., 2025; Tang et al., 2023) have demonstrated the superior unsupervised representation learning abilities of diffusion models, where the diffusion representation is extracted from the bottleneck layer at certain timesteps of the learned denoiser. These works leverage diffusion representations for various downstream tasks, including classification, segmentation, and image correspondence, often achieving performance comparable to or even exceeding established self-supervised learning (SSL) methods. In parallel, regularizing diffusion representations using powerful self-supervised learners, such as DINOv2 (Oquab et al., 2024) and MAE (He et al., 2022), can significantly improve the training efficiency and generation quality of diffusion models (Yu et al., 2025; Singh et al., 2026). This highlights the strong interplay between representation learning and generative modeling within these paradigms.

Despite these advances, the representation learning paradigms of diffusion models and traditional SSL remain quite different. Diffusion models are trained with a denoising objective, recovering clean signals from Gaussian corrupted inputs, whereas most SSL methods (Bardes et al., 2022; Chen et al., 2020; 2024; Zbontar et al., 2021; Oquab et al., 2024) are explicitly designed to enforce invariance to data augmentations while preserving a rich, diverse embedding space. The distinction between training objectives raises natural questions about diffusion representation spaces: to what extent do they implicitly capture the beneficial characteristics directly optimized in SSL, and how do these properties evolve across varying noise levels and throughout the learning dynamics?

Bridging diffusion representations with those studied in standard SSL can help clarify how diffusion training shapes features and how these features might be further improved to benefit diffusion models. Moreover, adopting a representation-centered viewpoint of diffusion models offers an intrinsic way to understand these systems: if diffusion models are indeed powerful representation learners, the properties of their learned representations should encode clear signatures of whether the model is capturing low-dimensional structure in image manifolds (Wang et al., 2024; Li et al., 2026b) rather than merely overfitting to the idiosyncratic details of the training data.

Moreover, this perspective is particularly helpful for understanding and monitoring the *generalizability* of diffusion models during training. Evaluating how well a model goes beyond memorizing the training set is practically difficult, especially in data-limited regimes where recent work has documented a distinct "early learning" phase (Li et al., 2023b; 2024a; Zhang et al., 2025; Baptista et al., 2025; Bonnaire et al., 2026): the model first learns to generalize and then gradually starts to memorize individual samples. In this regard, Stein et al. (2023) showed that standard generation metrics such as Fréchet distance (FID) are not reliable memorization detectors, while exhaustive nearest neighbor tests (Pizzi et al., 2022; Zhang et al., 2024) rely on large numbers of generated samples and are often expensive to run. By importing insights from SSL and evaluating diffusion models through the geometry of their learned representations, we obtain intrinsic, training time signals that track the quality of the representation space. These signals offer a practical way to gauge generalized generation quality as training progresses and to identify good stopping points before overfitting dominates.

**Summary of contributions.** In this work we revisit diffusion models from a self-supervised representation learning perspective. Guided by classic SSL principles, we focus on two properties of internal features: *representation invariance*, reflecting the stability of shared content across random perturbations, and *representation expansion*, reflecting how well the features spread out in the available embedding space. We capture these two properties with a new intrinsic metric, the Invariant Contamination Ratio (ICR), which measures how much augmentation and noise sensitive variation contaminates the stable part of the representation space. To construct ICR, we introduce an *invariance–residual* decomposition of diffusion features that separates perturbation invariant structure from residual variation. Because ICR is label-free and can be computed entirely from training features, it can be monitored throughout training and across noise levels. Empirically, we find that ICR provides a reliable proxy for noise level dependent downstream representation performance and cleanly separates generalization

from memorization during training in data-limited regimes. In summary, our main contributions are as follows:

- **A representation-based evaluation metric for diffusion models.** We introduce an invariance–residual decomposition of diffusion representations and from it define the Invariant Contamination Ratio (ICR), a single label-free scalar that measures how much of the representation space is occupied by augmentation and noise-sensitive variation rather than stable structure. ICR can be computed solely from training features without labels or external networks.

- **Finding the optimal representation across noise levels.** On standard image benchmarks, we show that the diffusion noise schedule admits an intermediate *semantic window* where ICR is minimized and linear classification accuracy is maximized. This gives a simple SSL-based rule to select noise scales that yield the strongest diffusion features for downstream tasks.

- **Tracking generalization and memorization in training dynamics.** By following ICR over training, we observe distinct learning phases: in data-rich regimes it decreases steadily with improving generative quality, while in data-limited regimes it exhibits a U-shaped early learning pattern that precedes the rise of memorization. Thus ICR provides a practical, label-free early stopping signal, and its decomposition reveals that feature expansion during training is driven by invariant structure when data are abundant and by residual variation when data are scarce.

## 2. Problem Setup

**Preliminaries on diffusion models.** Diffusion models define a forward process that gradually perturbs data $x_0 \sim p_{\text{data}}$ toward a Gaussian distribution via the stochastic differential equation

$$\mathrm{d}x_t = f(t)x_t\,\mathrm{d}t + g(t)\,\mathrm{d}w_t, \quad t \in [0,1],$$

where $f$ and $g$ are scalar functions and $\{w_t\}$ is a standard Wiener process. Let $p_t$ denote the density of $x_t$ and note that $p_0 = p_{\text{data}}$. For simplicity, we consider the variance preserving setting $x_t = x_0 + \sigma_t\epsilon$ with $\epsilon \sim \mathcal{N}(0, I)$. The reverse time process that maps noise back to data uses the score $\nabla \log p_t(x_t)$ and is given by the reverse SDE (Anderson, 1982)

$$\mathrm{d}x_t = \big(f(t)x_t - g^2(t)\nabla \log p_t(x_t)\big)\mathrm{d}t + g(t)\,\mathrm{d}\bar{w}_t,$$

where $\{\bar{w}_t\}$ is an independent Wiener process. This enables diffusion models to generate new samples from the underlying data distribution $p_{\text{data}}$ by initializing from pure Gaussian noise and iteratively denoising via the score function.

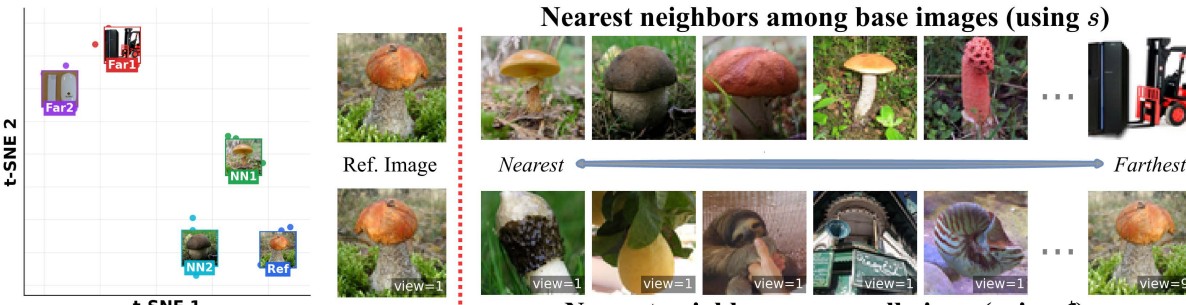

*Figure 1.* **Nearest neighbors of invariant and residual components on ImageNet** $64 \times 64$. We use a pretrained EDM diffusion model and, for each ImageNet training image, sample 9 augmented views and extract bottleneck representations. **Left:** t-SNE visualization of invariant representations $s$; views of the same base image form tight clusters, and the marked nearest and farthest examples reflect their relative positions in this space. **Top right:** For a reference image (left), nearest and farthest neighbors among the base images are selected using cosine similarity on $s$. **Bottom right:** For a reference augmented view (left), nearest and farthest neighbors across all augmented views are selected using cosine similarity on $\xi$. Neighbors based on $s$ tend to be semantically similar to the query, whereas neighbors based on $\xi$ often appear semantically unrelated.

**Training loss of diffusion models.** Modern diffusion models are typically trained to approximate the score function $\nabla \log p_t(\boldsymbol{x}_t)$. By Tweedie's formula ([Efron, 2011]),

$$\mathbb{E}\left[\boldsymbol{x}_0 \mid \boldsymbol{x}_t\right] = \boldsymbol{x}_t + \sigma_t^2 \nabla \log p_t(\boldsymbol{x}_t), \quad (1)$$

this is equivalent to learning the posterior mean $\mathbb{E}[\boldsymbol{x}_0 \mid \boldsymbol{x}_t]$ via a denoising autoencoder $\boldsymbol{x}_{\boldsymbol{\theta}}(\boldsymbol{x}_t, t)$ ([Chen et al., 2025]; [Xiang et al., 2023]; [Kadkhodaie et al., 2024]). Concretely, we minimize the weighted denoising loss

$$\min_{\boldsymbol{\theta}} \sum_{i=1}^{N} \int_0^1 \lambda_t \, \mathbb{E}_{\boldsymbol{\epsilon}} \left[ \left\| \boldsymbol{x}_{\boldsymbol{\theta}}(\boldsymbol{x}_t^{(i)}, t) - \boldsymbol{x}_0^{(i)} \right\|^2 \right] \mathrm{d}t, \quad (2)$$

where $\boldsymbol{x}_0^{(i)} \overset{i.i.d.}{\sim} p_{\text{data}}$ for $i = 1, \dots, N$ and $\lambda_t$ weights different noise levels.

**Layer selection for extracting representations.** We freeze the diffusion backbone and extract representations from the layer that gives the strongest downstream performance according to [Xiang et al. (2023)]. In practice, this corresponds to a layer near the bottleneck of the U-Net architecture ([Ronneberger et al., 2015]; [Karras et al., 2022]) and the middle transformer block of SiT ([Ma et al., 2024]).

## 3. A Representation-Level Evaluation Framework from Self-Supervised Principles

As discussed in the introduction, recent work shows that diffusion models can act as strong self supervised representation learners, supporting competitive performance across various downstream tasks ([Baranchuk et al., 2022]; [Xiang et al., 2023]; [Chen et al., 2025]; [Fuest et al., 2026]). However, their training paradigm differs markedly from standard self-supervised learning (SSL): while most SSL methods rely on explicit contrastive or predictive objectives to shape the embedding space ([Chen et al., 2020]; [He et al., 2020]; [Grill et al., 2020]; [Oquab et al., 2024]), diffusion models are trained with a denoising objective that reconstructs clean signals from Gaussian corrupted inputs.

This difference raises a fundamental question:

> To what extent does the denoising objective naturally satisfy the geometric properties of "good" representations sought in the SSL literature?

In diffusion models, the internal representations are high-dimensional and evolve across different noise scales, making it non-trivial to separate stable information from idiosyncratic variation. To evaluate this, we propose an evaluation metric rooted in the principles of modern image-based SSL.

### 3.1. What Makes a Good Representation in SSL?

Modern SSL methods ([Oord et al., 2018]; [Wang & Isola, 2020]; [Chen et al., 2020]; [Bardes et al., 2022]; [Zbontar et al., 2021]; [Oquab et al., 2024]) are often built around two complementary principles:

- **Representation invariance**: Representations extracted from different stochastic perturbations of a sample are encouraged to remain stable in the embedding space.

- **Representation expansion**: Representations should maintain a rich, spread-out structure across different images. The embedding distribution should avoid dimensional collapse and utilize many directions in the representation space to preserve unique image identities.

In the rest of this work, we use these principles to track how diffusion representations evolve across noise levels and training. We introduce a simple decomposition that splits each representation into a *perturbation invariant component*, stable across noisy and augmented views, and a

*residual component* that captures variation induced by these perturbations.

## 3.2. A Geometric Framework for Representation Evaluation

Guided by the SSL principles in Section 3.1, we seek a representation-level metric that can be monitored across noise levels and training, and that reflects how much of the active representation space is devoted to perturbation invariant structure. Informally, we want this diagnostic to (i) measure *relative* invariance rather than an absolute distance scale, (ii) be robust to overall representation expansion during training, and (iii) remain label-free and efficiently computable from representations. A natural starting point is the Alignment and Uniformity criteria of Wang & Isola (2020), which have been highly successful in characterizing contrastive SSL encoders. However, Alignment is an absolute squared distance between two augmented views of the same image and grows when representations take more directions, so it can increase even when the representation becomes more semantically stable, while Uniformity only measures how spread out representations are and does not distinguish invariant structure from augmentation-sensitive noise. (see Appendix B.3 for details).

These limitations motivate a different construction that explicitly separates invariant information from view-specific variation. Instead of working directly with distances between raw representations, we first decompose the representation space into stable and varying components, then leverage the spectral properties of these components to derive a summary metric for representations.

**Invariant and residual decomposition in representation space.** For each training image $x_0 \sim p_{\text{data}}$, let $a \sim \mathcal{A}$ denote a random perturbation encompassing both standard semantics-preserving transformations (Chen et al., 2020; He et al., 2020) and the additive Gaussian noise $\epsilon \sim \mathcal{N}(0, \sigma_t^2 I)$ injected by the diffusion objective[1]. Let $h(\cdot) \in \mathbb{R}^d$ be the representation extracted from a fixed layer of the diffusion model, and consider the random representation $h(a(x_0))$ induced by the stochasticity of the perturbation $a$. We decompose this representation into its conditional mean and a residual:

$$s(x_0) := \mathbb{E}_a\big[h(a(x_0)) \mid x_0\big],$$
$$\xi(a, x_0) := h(a(x_0)) - s(x_0), \tag{3}$$

yielding the additive form $h(a(x_0)) = s(x_0) + \xi(a, x_0)$. In this decomposition, $s(x_0)$ is the *invariant component*, which filters out transient variations to capture attributes resilient to corruption. Conversely, $\xi(a, x_0)$ is the *residual*

---

[1]Specifically, we first apply augmentations to $x_0$ and then add Gaussian noise to the augmented view to obtain $a(x_0)$.

*component* capturing the specific idiosyncratic variations of a single noisy view.

Figure 1 provides empirical support for this interpretation: multiple perturbed views of a single image form a cluster in the representation space centered at $s(x_0)$. Notably, nearest-neighbor searches based on $s(x_0)$ retrieve semantically related images, whereas neighbors based solely on the residual $\xi(a, x_0)$ appear visually unrelated and lack shared category structure.

Since $\mathbb{E}[\xi \mid x_0] = 0$, the law of total covariance implies that the total representation covariance $\Sigma_h$ decomposes into two components:

$$\Sigma_h = \Sigma_s + \Sigma_\xi,$$
$$\text{where} \quad \Sigma_s := \text{Cov}_{x_0}(s), \quad \Sigma_\xi := \text{Cov}_{x_0,a}(\xi).$$

This decomposition allows us to translate core SSL principles into geometric properties of the representation space:

- **Representation expansion** refers to the growth of the total covariance $\Sigma_h = \Sigma_s + \Sigma_\xi$, which characterizes how features spread in the representation space across data samples. More specifically, it describes the extent to which the feature covariance occupies multiple directions, i.e., whether the representation is low-rank or broadly distributed in the embedding space. In practice, we track its trace, $\text{Tr}(\Sigma_h)$, which measures the total variance (energy) of the representation and reflects how much of the high-dimensional feature space the model utilizes for distinguishing between different images and their perturbed views.

- **Representation invariance** is measured by the relative dominance of $\Sigma_s$ over $\Sigma_\xi$. Unlike traditional SSL, diffusion models require a noise-dependent balance: at low noise levels, a larger residual $\Sigma_\xi$ is necessary for reconstructing fine-grained pixel details, whereas at intermediate noise levels, a higher degree of invariance is desired to capture stable semantic structures.

**Fisher directions and invariant signal-to-noise ratios.** Given the invariant and residual covariances $(\Sigma_s, \Sigma_\xi)$, we consider the generalized eigenproblem (Fukunaga, 1990):

$$\Sigma_s v_i = \lambda_i \Sigma_\xi v_i, \qquad \lambda_1 \geq \lambda_2 \geq \cdots \geq \lambda_d \geq 0,$$

with eigenvectors $v_i$ orthogonal in the $\Sigma_\xi$ inner product. In practice, these matrices are estimated empirically across the training dataset, where $\Sigma_\xi$ aggregates residual variations within each sample and $\Sigma_s$ captures the spread of invariant identities across all samples. Each $v_i$ represents a direction in representation space that is optimized to maximize the

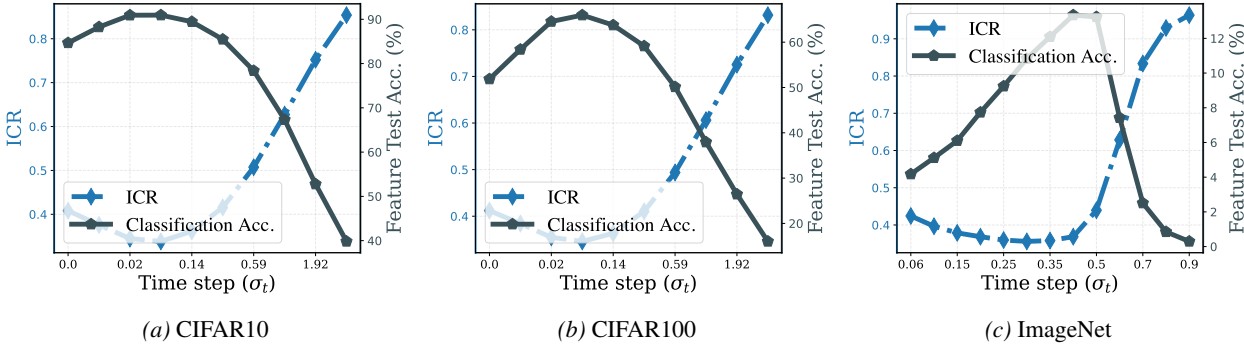

*Figure 2.* **Correspondence between ICR and classification accuracy across noise levels.** For each pretrained backbone (EDM (Karras et al., 2022) on CIFAR10 and CIFAR100, SiT-XL/2 (Ma et al., 2024) on ImageNet), we extract bottleneck representations at multiple noise levels $\sigma_t$. At each $\sigma_t$, we estimate ICR (blue) using a subset of training representations and train a classifier on the full training representations, reporting accuracy on the test set (slate). Across datasets, the noise levels that minimize ICR coincide with those that maximize classification accuracy.

ratio of invariant signal energy to residual variation. Specifically, the corresponding eigenvalue $\lambda_i$ admits the Rayleigh quotient (Horn & Johnson, 2012) representation[2]:

$$\lambda_i = \max_{\boldsymbol{v} \neq 0, \; \boldsymbol{v} \perp_{\boldsymbol{\Sigma}_\xi} \{\boldsymbol{v}_1, \ldots, \boldsymbol{v}_{i-1}\}} \frac{\boldsymbol{v}^\top \boldsymbol{\Sigma}_s \boldsymbol{v}}{\boldsymbol{v}^\top \boldsymbol{\Sigma}_\xi \boldsymbol{v}}. \quad (4)$$

In this framework, a generalized eigenvalue $\lambda_i$ measures the *invariant signal-to-noise ratio* along the corresponding Fisher direction (Fisher, 1936) $\boldsymbol{v}_i$: it compares the variance of the stable invariant component $\boldsymbol{v}^\top \boldsymbol{\Sigma}_s \boldsymbol{v}$ to the residual variance $\boldsymbol{v}^\top \boldsymbol{\Sigma}_\xi \boldsymbol{v}$ in that specific direction. This follows the same generalized eigenstructure as classical Fisher Linear Discriminant Analysis (Fisher, 1936), where $\boldsymbol{\Sigma}_s$ and $\boldsymbol{\Sigma}_\xi$ play roles analogous to between-class and within-class covariances, respectively, with each individual image effectively acting as its own class.

**Invariant Contamination Ratio (ICR).** The collection of generalized eigenvalues $\{\lambda_1, \ldots, \lambda_d\}$ provides a directional profile of how strongly invariant structures dominate residual variations. To summarize this into a single, trackable scalar that quantifies the health of the representation, we define the *Invariant Contamination Ratio* (ICR):

$$\mathrm{ICR} := \frac{1}{1 + \frac{1}{d} \sum_{i=1}^{d} \lambda_i}. \quad (5)$$

The term $\frac{1}{d} \sum \lambda_i$ represents the average invariant signal-to-noise ratio across all Fisher directions. When the invariant component $\boldsymbol{s}$ dominates the residual $\boldsymbol{\xi}$ across the majority of directions, this average is large, resulting in a low ICR. Conversely, as residual variation (or "contamination") increases and begins to occupy a substantial portion of the representation space, the ICR approaches 1.[3] We note that

---

[2]We assume $\boldsymbol{\Sigma}_\xi \succ \boldsymbol{0}$; in practice we can add a small $\tau > 0$ and replace $\boldsymbol{\Sigma}_\xi$ by $\boldsymbol{\Sigma}_\xi + \tau \boldsymbol{I}$ to ensure invertibility.

[3]This choice matches the convention of generative metrics like FID, where lower values indicate a "cleaner" and more robust

in practice, we estimate $\boldsymbol{\Sigma}_s$ and $\boldsymbol{\Sigma}_\xi$ from as few as two augmentations per image and a subset of training representations; implementation details are given in Section B.6.

# 4. ICR **across Noise Levels: a Semantic Window for Representation Learning**

The denoising objective of diffusion models is inherently multiscale: for each clean image $\boldsymbol{x}_0$ the model sees a family of corrupted inputs $\{\boldsymbol{x}_t\}$ indexed by the noise level $\sigma_t$, and thus induces a family of representations $\boldsymbol{h}_t(a(\boldsymbol{x}_0))$. In this section we use ICR to answer two questions:

- *How does relative invariance vary across the diffusion noise schedule?*

- *Can this internal measure predict which noise levels yield the best downstream representations?*

**ICR predicts classification performance and reveals a semantic window.** We start from pretrained diffusion backbones on CIFAR datasets (Krizhevsky, 2009) and ImageNet (Deng et al., 2009). For each noise level $\sigma_t$, we estimate $\mathrm{ICR}(\sigma_t)$ using a subset of training representations extracted from inputs corrupted at noise level $\sigma_t$, and train a linear classifier on the full training representations at the same $\sigma_t$. Figure 2 plots ICR and test accuracy as functions of $\sigma_t$.

Across all datasets we observe a striking alignment: the ICR curve is U-shaped and attains a clear minimum at an intermediate noise level, while classification accuracy peaks in exactly the same range. We refer to this range as a *semantic window*: at very low noise, representations are too tied to fine-grained, augmentation-specific details; at very high noise, representations collapse toward noise; in between, relative invariance is strongest and the model uses its representation space in a way that is most useful

---

representation.

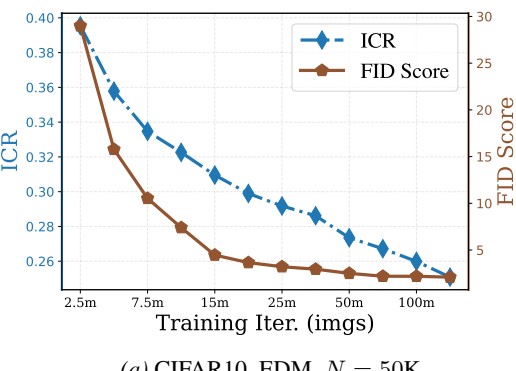

*(a)* CIFAR10, EDM, $N = 50K$

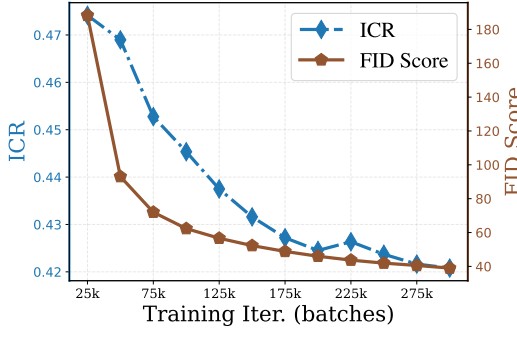

*(b)* ImageNet-256, SiT-B/2, $N = 1.28M$

*Figure 3.* ICR **and FID dynamics in data rich diffusion training.** We monitor generative performance (via FID) and ICR for EDM and SiT-B/2 based diffusion models trained on the full CIFAR10 and ImageNet datasets as training progresses. Both ICR (blue) and FID (brown) exhibit a monotonically decreasing trend, indicating improving internal representation invariance and sample quality over the course of training.

for downstream tasks. Importantly, ICR is computed in a label-free way from training representations alone, yet it accurately predicts which noise scales will deliver the best linear probe performance. We further verify the robustness of this observation in the data-limited setting in Section B.2 and observe a consistent trend.

These observations are consistent with prior work (Wang & Vastola, 2023; Li et al., 2026b) showing that diffusion sampling exhibits a coarse-to-fine transition, with different noise levels capturing structure at different granularities. They also suggest that our augmentation-based evaluation developed from self-supervised principles extend naturally to diffusion models, providing a simple bridge between these two frameworks.

Motivated by this semantic window, in the next section we fix a representative intermediate noise level $\sigma^\star$ near the ICR minimum and study how the representation space evolves over the course of training at this scale, relating the dynamics of invariance to generative quality and memorization.

## 5. Invariance and Expansion over Training: from Generalization to Memorization

In the previous section we examined how ICR varies across the diffusion noise schedule and identified an intermediate *semantic window* where downstream representations are strongest. We now fix a representative noise scale $\sigma^\star$ in this window and track how the representation space evolves over the course of training.

Our analysis proceeds in three steps. First, in the data-rich regime, we show that ICR closely tracks FID during training, linking improvements in generative quality to changes in the internal feature geometry. Second, in the data-limited regime, we observe an early learning phenomenon in the representation space: ICR exhibits a clear U-shaped trajectory and its minimum aligns with the onset of memorization. Finally, we examine trace-level statistics of the invariant and

residual covariances to understand how feature expansion is allocated between invariant and residual components in the two regimes.

### 5.1. ICR **Tracks FID in the Data-Rich Regime**

We first consider the data-rich regime, where the diffusion model is trained on the full training set. At each checkpoint we compute ICR from training features and evaluate the Fréchet Inception Distance (FID) between generated samples and the real data distribution. Figure 3 illustrates the resulting trajectories as functions of training progress.

In this setting, ICR and FID exhibit a strong positive correlation: both demonstrate a monotonically decreasing trend. This alignment reflects an intuitive connection between generative quality and representation geometry. FID (Heusel et al., 2017) (and related Fréchet-based metrics such as $FD_{DINOv2}$) measures the distance between generated and real distributions in an external semantic feature space (Szegedy et al., 2015; Oquab et al., 2024), quantifying how well the model captures the data manifold. In contrast, ICR quantifies the "purity" of the internal diffusion representation by measuring the ratio of augmentation-sensitive residual energy to stable invariant signal.

The simultaneous improvement in FID and ICR suggests that the improvement in generative ability is directly reflected in the refinement of the representation space. As the model better approximates $p_{\text{data}}$, its internal features shift from capturing transient, view-specific noise toward a stable, low-dimensional image structure. In our framework, this manifests as a higher signal-to-noise ratio in the Fisher directions, where the invariant component $\Sigma_s$ increasingly dominates the residual variation $\Sigma_\xi$.

### 5.2. Early Learning in the Representation Space Under Limited Data

In this subsection, we investigate representation dynamics in the limited data regime. In this setting, recent studies (Li

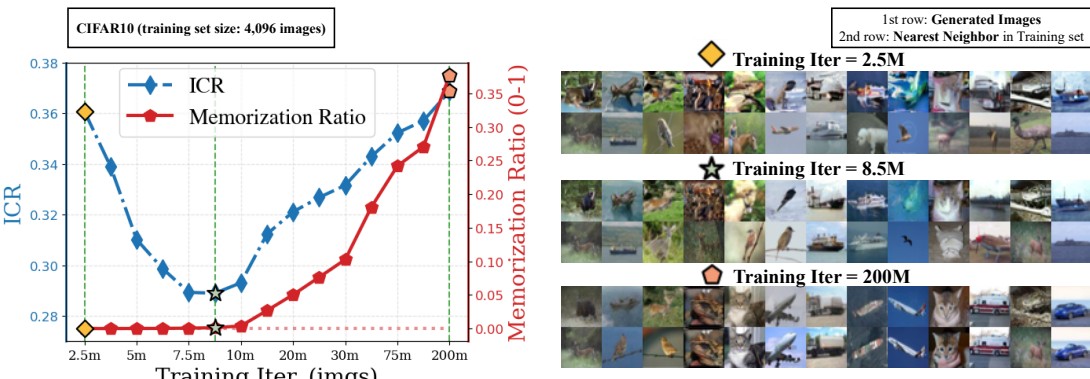

*Figure 4.* **ICR as an early signal of memorization in data limited diffusion training (CIFAR10).** We evaluate an EDM-based diffusion model trained on a subset of CIFAR10 (4096 images). **Left:** ICR (blue) follows a clear U shaped trajectory as training progresses, while the memorization ratio (red) remains near zero early on and begins to rise only after the ICR minimum. **Right:** Qualitative inspection at 2.5M, 8.5M, and 200M training images seen. Generated samples (top) and their nearest training neighbors (bottom) show that visual quality initially improves, but eventually the model begins to memorize individual training images, in line with the ICR curve.

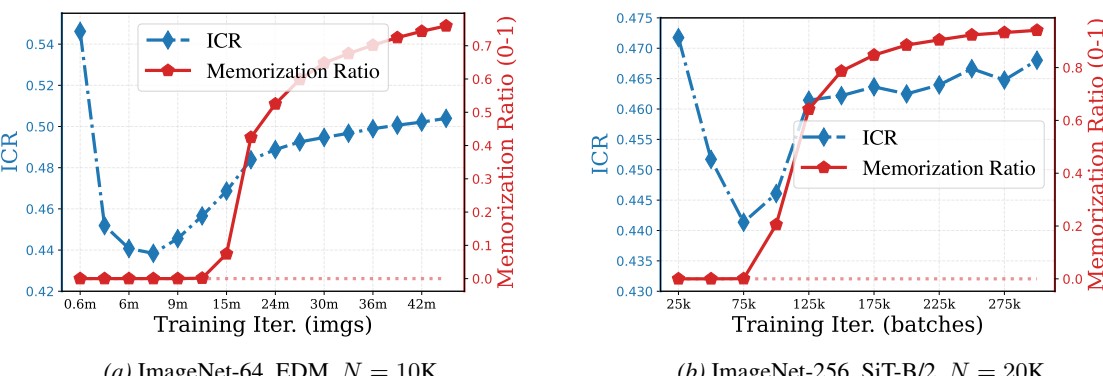

*(a)* ImageNet-64, EDM, $N = 10K$        *(b)* ImageNet-256, SiT-B/2, $N = 20K$

*Figure 5.* **ICR dynamics consistently anticipate memorization across large-scale datasets.** We repeat the analysis of ICR (blue) and memorization ratio (red) on ImageNet in data limited settings. **(a)** EDM trained on a 10K image subset of ImageNet $64 \times 64$. **(b)** SiT-B/2 diffusion model trained on a 20K image subset of ImageNet $256 \times 256$. In both cases, ICR dips and then rises before the memorization ratio increases, mirroring the behavior on CIFAR10 experiments (Figure 4).

et al., 2023b; 2024a; Baptista et al., 2025; Bonnaire et al., 2026) have demonstrated an early learning phenomenon: image generation quality first improves and the model generalizes well in an initial phase of training, before eventually deteriorating as the model begins to memorize.

To examine whether the representation space undergoes a similar early learning trajectory in data-limited settings, we track ICR and a memorization ratio (Pizzi et al., 2022; Zhang et al., 2024) across training. Concretely, we train an EDM (Karras et al., 2022) model on a subset of CIFAR10 with $N = 4096$ images, an EDM on a 10K subset of ImageNet $64 \times 64$, and a SiT-B/2 (Ma et al., 2024) model on a 20K subset of ImageNet $256 \times 256$, and for each setting report ICR computed on training features together with the memorization ratio.[4]

In these experiments, as reported in Figures 4 and 5, ICR follows a clear U-shaped curve, in sharp contrast to the data-rich case in Figure 3. This indicates that feature invariance in the limited data regime also exhibits an early learning pattern: it improves during the initial phase of training and then degrades as training continues.

We moreover observe that the memorization ratio remains essentially zero around the ICR minimum and begins to increase only afterward. This suggests that the onset of memorization is reflected directly in the representation space: beyond the point where ICR is minimized, the model increasingly fits sample-specific idiosyncrasies (Zhang et al., 2026) rather than shared, stable semantic structure.

This transition in the representation is also visible in Figure 6, where we use $s$ to find nearest neighbors across

---

[4]The memorization ratio is computed by generating 10K images, extracting features for generated and training images using an external encoder (Pizzi et al., 2022), and, for each generated image,

checking whether its maximum cosine similarity to any training image exceeds 0.6. Such samples are counted as memorized.

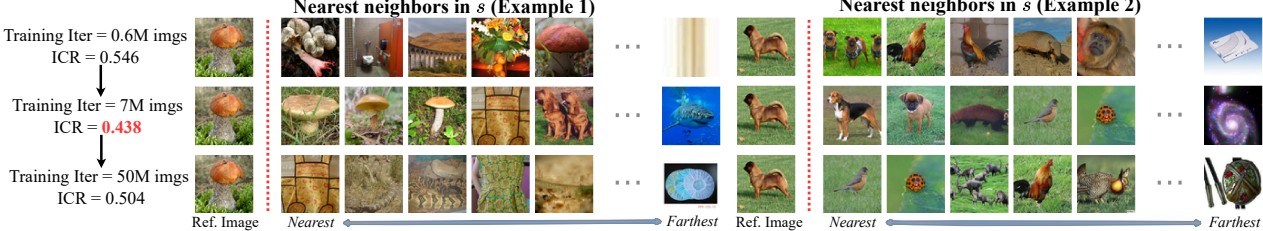

Figure 6. **Nearest neighbors of invariant components throughout limited data training.** We visualize nearest neighbors in $s$ as in Figure 1. The neighbors qualitatively track ICR and the model's generalization: in the first row (early training, near initialization), ICR is large and neighbors are not semantically meaningful; in the second row (intermediate training), ICR is smallest and neighbors are reasonable; in the third row (severe overfitting), ICR increases again and neighbor quality degrades.

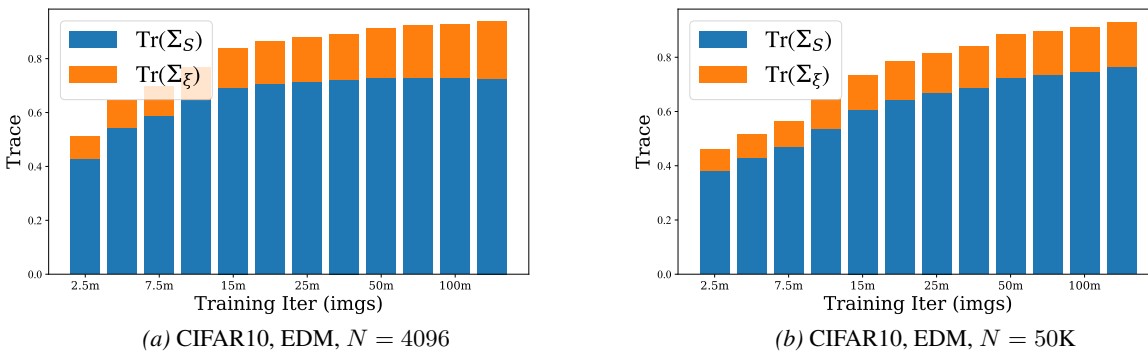

$(a)$ CIFAR10, EDM, $N = 4096$          $(b)$ CIFAR10, EDM, $N = 50K$

Figure 7. **How feature expansion differs in data-limited and data-rich diffusion training.** We train two EDM-based diffusion models on CIFAR10 with different training set sizes and track the traces of the invariant and residual covariances over training (as labeled).

the training trajectory in the data-limited case. During the middle phase of training, when ICR is small, the nearest neighbors are semantically close to the query, whereas in the early under-trained phase and the late heavily memorizing phase, where ICR is larger, the retrieved neighbors are noticeably less meaningful.

Finally, we note that the ICR values reported here are computed purely from training features and do not require generation or an external evaluation network. Taken together with the previous subsection, this suggests that ICR can serve as a reliable "generalized" generation metric that is monitorable during training without sampling. In the data-rich regime, it closely tracks improvements in generative quality in a way that parallels FID. In the data limited regime, where prior work has pointed out that FID is not entirely trustworthy for detecting memorization (Stein et al., 2023), ICR can act as a coarse, easy-to-monitor early stopping signal, complementary to standard generation-based metrics such as the memorization ratio (Pizzi et al., 2022).

### 5.3. How Feature Expansion Differs in Data-Rich and Data-Limited Regimes

In the previous subsections, we highlighted that the relative feature invariance, as measured by ICR, follows very different trajectories in the data-rich and data-limited regimes. However, ICR is a *relative* quantity: it summarizes how much residual variance contaminates invariant directions,

but does not reveal how the *total* representation energy evolves. In particular, when ICR increases in the late stage of training under limited data, it is not clear whether this reflects an overall shrinkage of the representation space, a reallocation of variance from invariant to residual components, or some combination of both. To disentangle these possibilities, we examine the traces of the invariant and residual covariances over training.

In Figure 7, we train two EDM-based diffusion models with different training set sizes to compare a data-limited setting (4096 images) with a data-abundant setting (50K images). Across both settings, the total feature energy, measured by $\mathrm{Tr}(\mathbf{\Sigma}_s) + \mathrm{Tr}(\mathbf{\Sigma}_\xi)$, increases steadily over training. The regimes differ, however, in how this growth is allocated between invariant and residual components. In the data-abundant case, $\mathrm{Tr}(\mathbf{\Sigma}_s)$ continues to increase throughout training, while $\mathrm{Tr}(\mathbf{\Sigma}_\xi)$ grows only mildly, indicating that additional feature capacity is predominantly devoted to invariant structure. In the data limited case, $\mathrm{Tr}(\mathbf{\Sigma}_s)$ rises initially but then saturates, whereas $\mathrm{Tr}(\mathbf{\Sigma}_\xi)$ continues to grow. This suggests that, once the limited semantic structure in the dataset has been largely extracted, further feature expansion is dominated by augmentation-sensitive residual variability. Consistent with this picture, ICR decreases with training in the data-abundant regime but exhibits a U-shaped trajectory under limited data, reflecting the late stage shift from invariant to residual energy.

# 6. Conclusion

In this work we revisit diffusion models from a self-supervised representation learning perspective. We introduce an invariance–residual decomposition of diffusion representations and the Invariant Contamination Ratio (ICR), a label-free metric that measures how much augmentation and noise-sensitive variation contaminates stable structure in the feature space. Using this framework, we demonstrate that diffusion noise levels admit a semantic window where ICR is minimized and downstream linear classification performance is maximized, providing a simple SSL-based way to identify the most informative denoising scales. Tracking ICR over training further reveals distinct learning phases: in data-rich regimes it decreases in tandem with improvements in generative quality, while in data-limited regimes it exhibits an early learning pattern that anticipates the onset of memorization. These findings suggest that diffusion models can be monitored and evaluated through their own representation space, providing intrinsic training-time signals that complement conventional generation-based metrics.

## Acknowledgment

XL, YJ, XL, LS, ZZ (UM) and QQ acknowledge support from NSF CAREER CCF-2143904, NSF CCF-2212066, NSF CCF-2212326, NSF IIS 2402950, ONR N000142512339, and Google Research Scholar and Google TPU Award. LS and QQ acknowledge support from DARPA HR00112520042. JZ and ZZ (OSU) acknowledge funding support from NSF IIS 2312840 and IIS 2402952. LS (UM) acknowledges funding support from NSF IIS 2435746. We also thank all the anonymous reviewers for their valuable suggestions and fruitful discussions.

## Impact Statement

This paper presents work whose goal is to advance the field of Machine Learning. There are many potential societal consequences of our work, none which we feel must be specifically highlighted here.

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

## A. Related Work

**Diffusion-based representation learning.** Many works treat a trained diffusion denoiser as a feature extractor and test its features on downstream tasks. These features work well for image classification (Xiang et al., 2023; Mukhopadhyay et al., 2023; Deja et al., 2023), segmentation (Baranchuk et al., 2022), correspondence (Zhang et al., 2023; Tang et al., 2023), and image editing (Shi et al., 2024), and recent surveys summarize this line of work (Fuest et al., 2026). Some articles also use diffusion models to generate augmentations and improve robustness under covariate shift (Sastry et al., 2024). Since representation quality often depends on the noise level used for feature extraction, several distillation and compression methods aim to reduce the need for expensive timestep search and to improve transfer (Yang & Wang, 2023; Li et al., 2023a; Stracke et al., 2025; Luo et al., 2023). Other work changes the training objective or the network to better combine generation and representation learning, for example, by adding new information-based losses or by building an explicit autoencoding structure (Mittal et al., 2023; Wang et al., 2023a; Hudson et al., 2024; Preechakul et al., 2022). (Han et al., 2025) further studies whether diffusion models learn hidden dependencies among image features. In contrast to methods that mainly aim to improve downstream transfer or generation, we study how diffusion features change across noise levels and across training, and we link this behavior to self-supervised principles.

**Representation dynamics and links to self-supervised learning.** Li et al. (2026b) study why diffusion representations often peak at an intermediate noise level and explain this unimodal behavior through a low-dimensional data model. They further show that this unimodal pattern disappears when diffusion models transition from generalization to memorization (Li et al., 2026b). More recently, Wang et al. (2026b) connect diffusion models and self-supervised learning through a shared perturbation-kernel perspective and propose a spectral alignment objective that improves diffusion training. This SSL perspective is also reflected in recent diffusion training methods such as REPA, which align diffusion representations with embeddings from pretrained self-supervised encoders (e.g., DINOv2) to improve generation quality and training efficiency (Oquab et al., 2024; Yu et al., 2025; Singh et al., 2026).

Our work differs from these lines in its goal. Rather than modifying the training objective or introducing an external teacher encoder, we study diffusion representations through principles inspired by self-supervised learning. In particular, our work is closely related to a line of research that develops label-free metrics for predicting downstream representation quality and guiding model selection, including metrics based on covariance spectrum decay, effective rank, and Fisher-style covariance decompositions (Agrawal et al., 2022; Garrido et al., 2023; Thilak et al., 2024). Similar to these works, we seek to evaluate representation quality directly from feature statistics without training downstream classifiers. However, unlike prior metrics designed primarily for representation selection in SSL, we leverage the invariant and residual decomposition induced by augmentations and diffusion noise to study diffusion-specific phenomena, including semantic windows, generation quality, memorization, and training dynamics.

**Memorization and generalization in diffusion models.** Our work contributes to the broad line of research aiming to understand memorization and generalization behaviors in diffusion models (Kadkhodaie et al., 2024; Zhang et al., 2024; Kamb & Ganguli, 2025; Shi et al., 2026; Zhang et al., 2025). A large body of work has studied memorization in diffusion models from the perspectives of model complexity and data quantity (Wang et al., 2024; Achilli et al., 2024; Bonnaire et al., 2026; Buchanan et al., 2026; Wang et al., 2026a), and has shown that memorization typically emerges after an initial generalization phase when diffusion models are trained with limited data (Li et al., 2024a; Wang & Pehlevan, 2026; Bonnaire et al., 2026; Favero et al., 2025). Other works seek to explain why diffusion models are able to recover the underlying score function from discrete empirical samples (Niedoba et al., 2025; Lukoianov et al., 2025). The generation and generalization behaviors across the reverse diffusion process have also been studied in (Biroli et al., 2024; Sclocchi et al., 2025). In parallel, Ambrogioni (2024) established an asymptotic equivalence between generative diffusion models and modern Hopfield networks (associative memory networks), and a subsequent work (Pham et al., 2025) leveraged this associative memory perspective to identify spurious samples that emerge as diffusion models transition from generalization to memorization.

## B. Additional Discussions & Experiments

### B.1. Component Dynamics Across the Noise Schedule

In Section 4, we show that there exists a strong correspondence between ICR and classification performance across noise levels. In this subsection, we also track the energy progression of the total variance of the invariant signal, $\mathrm{Tr}(\boldsymbol{\Sigma}_s)$, and the residual variation, $\mathrm{Tr}(\boldsymbol{\Sigma}_\xi)$.

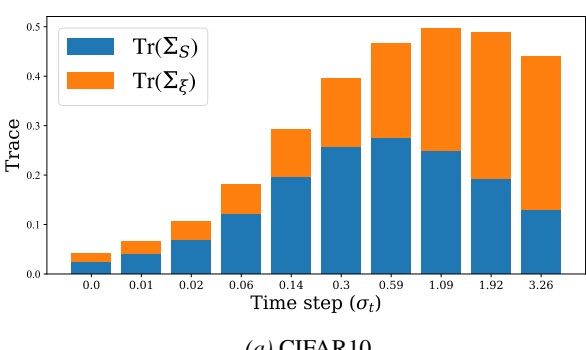

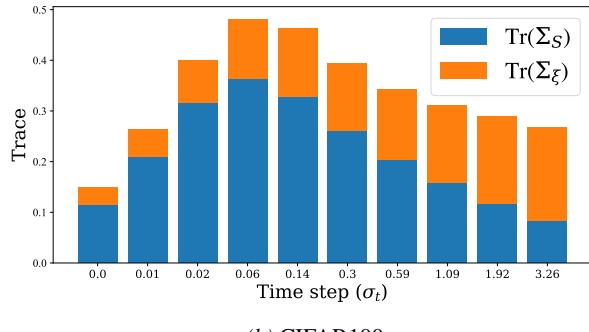

*(a)* CIFAR10         *(b)* CIFAR100

*Figure 8.* **Invariant and residual energy across diffusion noise levels.** For pretrained EDM models on CIFAR10 and CIFAR100 in the data-rich regime, we plot the traces of the invariant and residual covariances, $\mathrm{Tr}(\mathbf{\Sigma}_s(\sigma_t))$ and $\mathrm{Tr}(\mathbf{\Sigma}_\xi(\sigma_t))$, as functions of the noise level $\sigma_t$. Invariant energy $\mathrm{Tr}(\mathbf{\Sigma}_s)$ increases from low noise, peaks at an intermediate scale, and then decreases in the high noise regime, whereas residual energy $\mathrm{Tr}(\mathbf{\Sigma}_\xi)$ grows monotonically with $\sigma_t$.

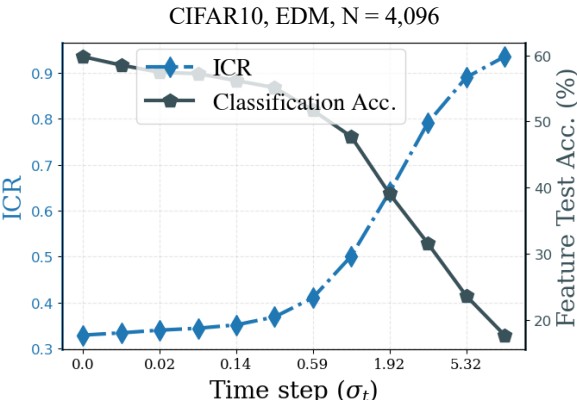

*Figure 9.* **Correspondence between ICR and classification accuracy across noise levels in the data-limited setting.** We study the behavior of ICR across noise levels in the data-limited setting under prolonged training. In this regime, classification accuracy no longer exhibits the unimodal trend observed in the generalization phase, and instead decreases monotonically as noise increases. In contrast, ICR increases monotonically and maintains a clear negative correlation with classification accuracy. This demonstrates that ICR continues to track representation quality even when classification accuracy no longer follows the typical generalization pattern.

As shown in Figure 8, $\mathrm{Tr}(\mathbf{\Sigma}_\xi)$ grows monotonically with the noise level $\sigma_t$, reflecting the increased reconstructive uncertainty inherent in high-noise denoising. Conversely, $\mathrm{Tr}(\mathbf{\Sigma}_s)$ exhibits a unimodal trajectory, peaking at intermediate scales. This suggests a "semantic window" where the model's invariant subspace is most expansive. Notably, peak classification accuracy (as shown in Figure 2) does not always correspond with the maximum of $\mathrm{Tr}(\mathbf{\Sigma}_s)$, but rather with the minimum of the ICR ratio. This indicates that representation utility is governed not by the absolute magnitude of the invariant signal, but by its strength relative to the contaminating residual variation.

### B.2. ICR and Classification Accuracy in the Data-limited Regime

In Figure 2, we show that ICR exhibits a strong negative correlation with classification accuracy across noise levels when diffusion models are trained using the full training set. Interestingly, the classification accuracy follows a unimodal trend as the noise level increases, a phenomenon previously observed and analyzed in (Xiang et al., 2023; Li et al., 2026b). Moreover, Li et al. (2026b) showed that this unimodal behavior is closely associated with the diffusion model correctly learning the underlying low-dimensional data distribution, and that it disappears once the model begins to memorize the training data, at which point the accuracy instead decreases monotonically with noise level.

To further test the robustness of the correlation between ICR and classification accuracy, we train an EDM model on a subset of 4,096 CIFAR10 images for a prolonged period until the model substantially overfits and memorizes the training data. We then evaluate both ICR and classification accuracy across noise levels at this checkpoint, and report the results in Figure 9.

As shown in the figure, classification accuracy no longer exhibits the unimodal trend observed during the generalization

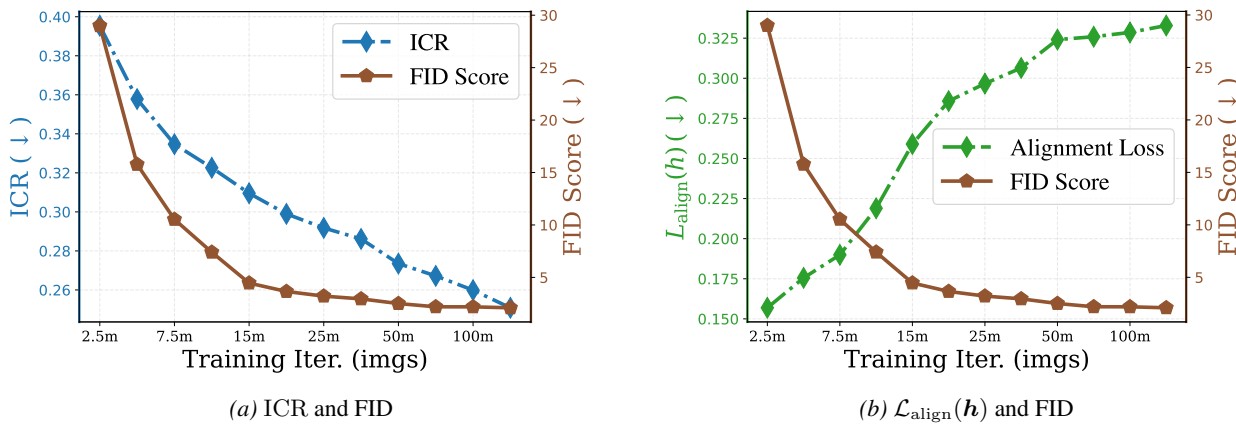

*(a)* ICR and FID  *(b)* $\mathcal{L}_{\mathrm{align}}(\boldsymbol{h})$ and FID

*Figure 10.* **Alignment versus ICR in data-rich diffusion training (CIFAR10, EDM).** We track FID together with ICR and the alignment loss $\mathcal{L}_{\mathrm{align}}$ over training on full CIFAR10. Both ICR (blue) and FID (brown) decrease monotonically, indicating improving representation invariance and generative quality, while $\mathcal{L}_{\mathrm{align}}$ (green) increases despite being a lower is better metric.

phase, and instead decreases monotonically as the noise level increases. Correspondingly, ICR exhibits a monotonic increasing trend across noise levels, further strengthening the correlation between ICR and representation quality even when the standard semantic window disappears due to memorization.

### B.3. Discussion on the Alignment and Uniformity Metrics (Wang & Isola, 2020)

Wang & Isola (2020) introduced two feature-level criteria for contrastive SSL encoders. In our notation, let $\boldsymbol{h}(a(\boldsymbol{x})) \in \mathbb{R}^d$ denote the feature of an augmented image, and let $(\boldsymbol{x}, \boldsymbol{x}')$ be a positive pair obtained from two augmentations of the same image. The *alignment* loss is

$$\mathcal{L}_{\mathrm{align}}(\boldsymbol{h}) \coloneqq \mathbb{E}_{(\boldsymbol{x}, \boldsymbol{x}')}\big[\|\boldsymbol{h}(a_1(\boldsymbol{x})) - \boldsymbol{h}(a_2(\boldsymbol{x}))\|_2^\alpha\big], \tag{6}$$

the expected squared distance between two views of the same sample. The *uniformity* loss is

$$\mathcal{L}_{\mathrm{uniform}}(\boldsymbol{h}; t) \coloneqq \log \mathbb{E}_{\boldsymbol{x}, \boldsymbol{y}}\Big[\exp\big(-t\|\boldsymbol{h}(a(\boldsymbol{x})) - \boldsymbol{h}(a'(\boldsymbol{y}))\|_2^2\big)\Big], \quad t > 0, \tag{7}$$

which encourages features to be spread out on the unit hypersphere. In the contrastive setting, encoders that achieve low alignment together with good uniformity tend to have strong downstream classification accuracy.

Our focus is slightly different. We are interested in how the representation space evolves across diffusion noise levels and along the training trajectory, and in particular in a *relative* notion of invariance that compares invariant structure to view specific variation while remaining stable under overall feature expansion. In this regime, the alignment loss becomes less informative. As shown in Figure 10, in the data-rich case the FID and ICR both decrease monotonically as training progresses, while $\mathcal{L}_{\mathrm{align}}$ continues to increase, suggesting worse alignment. This apparent disagreement is largely due to the fact that $\mathcal{L}_{\mathrm{align}}$ is an absolute squared distance: during diffusion training the overall feature variance grows (see Figure 7), so alignment can increase even when the relative invariant structure is improving.

### B.4. Discussion on the Class Separation and Silhouette Score Metrics

Besides the alignment and uniformity metrics discussed in (Wang & Isola, 2020), we additionally discuss two other representation metrics here: the class separation metric and the Silhouette score.

**Class separation** is a metric proposed in (Kornblith et al., 2021), which measures the within-class variation of representations relative to the overall variation.[5] Formally, based on cosine distances between normalized features, let $\boldsymbol{h}_{k,m}$ denote the representation of sample $m$ from class $k$, where $K$ is the number of classes and $N_k$ is the number of samples in class $k$. The class separation score $R^2$ is defined as

---

[5] The concept of maximizing separation between classes has recently been studied in improving diffusion model generation without classifier-free guidance (Li et al., 2026a).

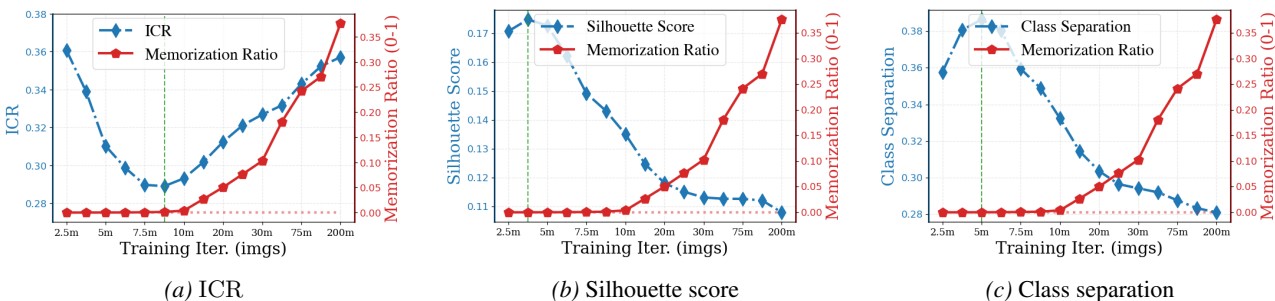

*(a)* ICR          *(b)* Silhouette score          *(c)* Class separation

*Figure 11.* ICR**, Silhouette score, and class separation in data-limited diffusion training (CIFAR10, EDM).** We revisit the experiment in Figure 4 (Training EDM on CIFAR10, with N=4,096 images) by incorporating two additional representation metrics: (i) the Silhouette Score, a partially unsupervised metric that relies on pseudo-labels (e.g., from k-means), and (ii) Class Separation, a supervised metric that depends on ground-truth labels. As shown in the figure, ICR exhibits a stronger and more consistent alignment with the memorization ratio, whereas the other two metrics saturate too early and fail to capture this trend as reliably.

$$R^2 = 1 - \frac{\bar{d}_{\text{within}}}{\bar{d}_{\text{total}}},$$

where

$$\bar{d}_{\text{within}} = \sum_{k=1}^{K} \sum_{m=1}^{N_k} \sum_{n=1}^{N_k} \frac{1 - \text{sim}(\boldsymbol{h}_{k,m}, \boldsymbol{h}_{k,n})}{K N_k^2},$$

$$\bar{d}_{\text{total}} = \sum_{j=1}^{K} \sum_{k=1}^{K} \sum_{m=1}^{N_j} \sum_{n=1}^{N_k} \frac{1 - \text{sim}(\boldsymbol{h}_{j,m}, \boldsymbol{h}_{k,n})}{K^2 N_j N_k}.$$

The metric was shown in (Kornblith et al., 2021) to correlate with transferability of ImageNet pre-trained models on downstream classification tasks. However, as the name suggests, the metric requires ground-truth class labels and is therefore not suitable for unsupervised settings. In addition, being a global statistic, it does not explicitly capture directional structure in the representation space as ICR does.

**Silhouette score** is a clustering-based metric that evaluates how well a representation separates data into groups (or classes when supervised labels are available). Given a feature representation $\boldsymbol{h}_i$, define

- $a(i)$: the average distance between $\boldsymbol{h}_i$ and samples within the same cluster,

- $b(i)$: the minimum average distance between $\boldsymbol{h}_i$ and samples in other clusters.

The Silhouette score is then defined as

$$s(i) = \frac{b(i) - a(i)}{\max(a(i), b(i))},$$

and the overall score is obtained by averaging $s(i)$ over all samples.

The Silhouette score measures the balance between intra-cluster compactness and inter-cluster separation, where larger values indicate better separation. However, this metric is not fully unsupervised, since it still requires cluster assignments, typically obtained through algorithms such as k-means, making it sensitive to hyperparameters such as the number of clusters. In addition, it relies on pairwise distances, which can become unstable in high-dimensional feature spaces, and similarly does not explicitly capture directional structure.

To compare these two metrics with ICR, we redo the experiments in Figure 4 and report the results in Figure 11. We observe that both the class separation score and the Silhouette score exhibit a unimodal trend during training, suggesting that they can partially capture the transition from learning the underlying data distribution to memorizing training samples. However, as shown in the figure, ICR exhibits a substantially stronger and more consistent alignment with the memorization ratio, whereas the other two metrics saturate much earlier and fail to reliably track the later-stage memorization behavior. We conjecture that this limitation is partly due to the global nature of these metrics.

## B.5. Connection to Neural Collapse

In the main text we showed that the generalized eigenvalues admit a direct SNR interpretation: for each Fisher direction $\boldsymbol{v}_i$, the generalized Rayleigh quotient equals $\lambda_i$, and

$$\text{ICR} = \frac{d}{d + \sum_{i=1}^{d} \lambda_i}, \qquad \sum_{i=1}^{d} \lambda_i = \text{Tr}\big(\boldsymbol{\Sigma}_\xi^{-1} \boldsymbol{\Sigma}_s\big).$$

This follows by writing $\widetilde{\boldsymbol{\Sigma}}_s \coloneqq \boldsymbol{\Sigma}_\xi^{-1/2} \boldsymbol{\Sigma}_s \boldsymbol{\Sigma}_\xi^{-1/2}$ and $\boldsymbol{u}_i = \boldsymbol{\Sigma}_\xi^{1/2} \boldsymbol{v}_i$, so that $\boldsymbol{u}_i$ are the eigenvectors of $\widetilde{\boldsymbol{\Sigma}}_s$ with eigenvalues $\lambda_i$ and $\text{Tr}(\boldsymbol{\Sigma}_\xi^{-1} \boldsymbol{\Sigma}_s) = \text{Tr}(\widetilde{\boldsymbol{\Sigma}}_s) = \sum_i \lambda_i$.

The trace form $\text{Tr}(\boldsymbol{\Sigma}_\xi^{-1} \boldsymbol{\Sigma}_s)$ could be linked to a familiar quantity in the Neural Collapse ($\mathcal{NC}$) literature (Papyan et al., 2020; Zhu et al., 2021). Neural Collapse refers to a phenomenon observed near the terminal phase of training in classification networks: penultimate features for each class collapse to a single mean vector, and these class means become maximally separated. A standard metric for quantifying this behavior is the $\mathcal{NC}_1$ score, defined for a $K$-class classifier as

$$\mathcal{NC}_1 = \frac{1}{K} \text{Tr}\big(\boldsymbol{\Sigma}_B^\dagger \boldsymbol{\Sigma}_W\big),$$

where

$$\boldsymbol{h}_G = \frac{1}{nK} \sum_{k=1}^{K} \sum_{i=1}^{n} \boldsymbol{h}_{k,i}, \qquad \bar{\boldsymbol{h}}_k = \frac{1}{n} \sum_{i=1}^{n} \boldsymbol{h}_{k,i}, \quad (1 \leq k \leq K),$$

$$\boldsymbol{\Sigma}_W \coloneqq \frac{1}{nK} \sum_{k=1}^{K} \sum_{i=1}^{n} \big(\boldsymbol{h}_{k,i} - \bar{\boldsymbol{h}}_k\big)\big(\boldsymbol{h}_{k,i} - \bar{\boldsymbol{h}}_k\big)^\top, \qquad \boldsymbol{\Sigma}_B \coloneqq \frac{1}{K} \sum_{k=1}^{K} \big(\bar{\boldsymbol{h}}_k - \boldsymbol{h}_G\big)\big(\bar{\boldsymbol{h}}_k - \boldsymbol{h}_G\big)^\top.$$

Here $\boldsymbol{\Sigma}_W$ and $\boldsymbol{\Sigma}_B$ denote the within-class and between-class covariance matrices of the penultimate features.

The design principle behind $\mathcal{NC}_1$ is closely related to ours: both metrics compare two covariance structures via a trace of a generalized eigenvalue type object, effectively measuring a signal-to-noise ratio in feature space. The key difference is that $\mathcal{NC}_1$ is label-based, contrasting between class and within-class variation, whereas ICR is built from a self-supervised principle that contrasts perturbation-invariant and perturbation-sensitive components without using labels.

Interestingly, several works have used $\mathcal{NC}_1$ and related quantities as metrics for assessing the transferability of pretrained discriminative models to downstream tasks (Galanti et al., 2022; Wang et al., 2023b; Li et al., 2024c; Wang et al., 2025). Our results suggest that a similar trace-based viewpoint extends naturally to diffusion models, and that a unified representation-based evaluation principle may be possible that covers both discriminative and generative settings through appropriate choices of "signal" and "noise" covariances.

## B.6. Technical Details on Calculating ICR

In Section 3.2, we introduced the formal definition of the metric ICR and argue that it can be efficiently estimated using only two augmented views and a subset of training features[6]. In this subsection, we briefly discuss alternative formulations of the metric, dive into more detail on the estimation of it and the robustness of the estimation regards different number of samples used.

---

[6]We use all generalized eigenvalues, rather than only the top ones, in order to summarize the invariant-to-residual ratio across the full representation space. Since the metric is directional, if the spectrum is sparse, it indicates that only a few directions carry strong invariant signal, while others are dominated by residual variation.

As noted in the main manuscript, the computation of ICR involves solving a generalized eigenvalue problem, one may argue that a simpler alternative is a trace-based statistic such as $\mathrm{Tr}(\boldsymbol{\Sigma}_\xi)/\mathrm{Tr}(\boldsymbol{\Sigma}_s)$, which aggregates all directions into a single global statistic. However, this aggregation loses directional information. For example, consider a representation where only a small number of directions carry strong invariant signal while the remaining directions are dominated by residual variation. In this case, the trace ratio averages over all directions and fails to reflect the presence of these highly informative directions. In contrast, the generalized eigenvalue formulation explicitly captures the signal-to-noise ratio along each direction and is therefore sensitive to such anisotropic structures.

**Two-view approximation and empirical estimation.** The conditional expectation over all augmentations and noise realizations is not available in practice. Following augmentation based self supervised learning, we approximate it using two independent views per image. For each $\boldsymbol{x}$, sample $a_1, a_2 \sim \mathcal{A}$ independently and set

$$\boldsymbol{h}_1 = \boldsymbol{h}(a_1(\boldsymbol{x}_0)), \qquad \boldsymbol{h}_2 = \boldsymbol{h}(a_2(\boldsymbol{x}_0)).$$

Under the decomposition above, this can be written as

$$\boldsymbol{h}_1 = \boldsymbol{s}(\boldsymbol{x}_0) + \boldsymbol{\xi}_1, \qquad \boldsymbol{h}_2 = \boldsymbol{s}(\boldsymbol{x}_0) + \boldsymbol{\xi}_2,$$

where $\boldsymbol{\xi}_v = \boldsymbol{\xi}(a_v, \boldsymbol{x}_0)$ are zero mean residuals that are conditionally uncorrelated across views given $\boldsymbol{x}$ and share the same covariance $\boldsymbol{\Sigma}_\xi$.

We construct effective semantic and nuisance covariances from the sum and difference of the two views. Define

$$\boldsymbol{t} := \tfrac{1}{2}(\boldsymbol{h}_1 + \boldsymbol{h}_2), \qquad \boldsymbol{d} := \boldsymbol{h}_1 - \boldsymbol{h}_2.$$

A direct covariance calculation under the assumptions above yields

$$\mathrm{Cov}(\boldsymbol{d}) = 2\,\boldsymbol{\Sigma}_\xi, \qquad \mathrm{Cov}(\boldsymbol{t}) = \boldsymbol{\Sigma}_s + \tfrac{1}{2}\,\boldsymbol{\Sigma}_\xi,$$

so that $\boldsymbol{\Sigma}_s$ and $\boldsymbol{\Sigma}_\xi$ can be recovered from the second moments of $\boldsymbol{t}$ and $\boldsymbol{d}$:

$$\boldsymbol{\Sigma}_\xi = \tfrac{1}{2}\,\mathrm{Cov}(\boldsymbol{d}), \qquad \boldsymbol{\Sigma}_s = \mathrm{Cov}(\boldsymbol{t}) - \tfrac{1}{4}\,\mathrm{Cov}(\boldsymbol{d}).$$

In our experiments we estimate $\mathrm{Cov}(\boldsymbol{t})$ and $\mathrm{Cov}(\boldsymbol{d})$ from paired augmentations within each dataset and obtain empirical covariances $\widehat{\boldsymbol{\Sigma}}_s$ and $\widehat{\boldsymbol{\Sigma}}_\xi$ via the same formulas.

**Estimating ICR from a subset of samples.** In practice, ICR is computed from empirical covariances $\widehat{\boldsymbol{\Sigma}}_s$ and $\widehat{\boldsymbol{\Sigma}}_\xi$ estimated from a finite set of features, so the estimate may deviate from its population value depending on the number of samples used. Under standard covariance concentration results for subgaussian features, the estimation error of $\widehat{\boldsymbol{\Sigma}}_s$ and $\widehat{\boldsymbol{\Sigma}}_\xi$ scales on the order of $\sqrt{d/N}$ in operator norm, where $d$ is the feature dimension and $N$ is the number of samples. Since ICR depends on these covariances only through the trace term $\mathrm{Tr}(\boldsymbol{\Sigma}_\xi^{-1}\boldsymbol{\Sigma}_s)$, we expect its finite sample estimate to be relatively stable once $N$ is moderately larger than $d$.

To verify this empirically, we perform a sample complexity study on CIFAR10. Fixing a pretrained EDM model and a representative noise level, we compute ICR using random subsets of training images with sizes

$$N \in \{16, 32, 64, 128, 256, 512, 1024, 2048, 4096, 8192, 16384, 32768, 50000\},$$

where 50,000 is the full training set size. As shown in Figure 12, the estimated ICR stabilizes quickly: even with a few hundred to a few thousand samples, the ICR trends are already very close to the full dataset estimate. This robustness justifies our use of relatively small subsets of training features to monitor ICR throughout the experiments in the main text.

**Sensitivity of ICR to augmentation design.** Since the computation of ICR relies on collecting representations under random augmentations, we further investigate its robustness to different augmentation strengths. Specifically, we consider five augmentation settings: Level 1 uses RandomCrop alone; level 2 adds horizontal flipping; level 3 further adds Color Jitter (the default setting used throughout the manuscript); level 4 additionally adds random rotation; and level 5 further adds CenterCrop. We denote these augmentation settings as Aug.1 through Aug.5.

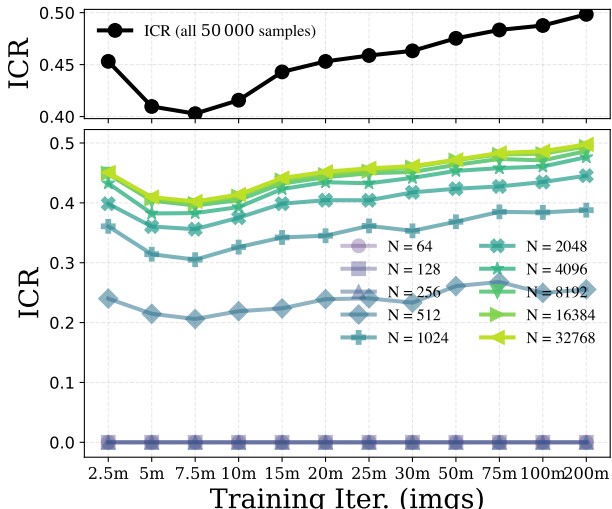

*Figure 12.* **Stability of** ICR **estimates under subsampling.** We evaluate ICR on CIFAR10 using a pretrained EDM model (4095 training samples) and a fixed noise level, varying the number of training samples used to estimate the covariances from $N = 16$ up to the full 50K images. As $N$ increases, the estimated ICR quickly showcase the similar trend close to the full data estimate.

Using these augmentation pipelines, we redo the experiments of ICR across noise levels in the data-rich setting and across training dynamics in the data-limited setting, and report the results in Figure 13. As shown in the figure, once a reasonably rich augmentation pipeline is used (e.g., random crop + flip + color jitter and stronger variants), the resulting ICR trends remain highly consistent, indicating that our findings are robust to the specific choice of augmentations.

In contrast, when overly weak augmentations are used (e.g., random crop alone), the behavior becomes noticeably less stable. We conjecture that this is because the invariant-residual decomposition requires sufficiently diverse perturbations in order to meaningfully separate invariant and residual components.

**Sensitivity of** ICR **to the choice of feature extraction layer.** We evaluate the sensitivity of ICR to layer selection. Specifically, we extract features from multiple layers around the middle of the diffusion model and compute ICR for each choice. The results in Figure 14 show that ICR exhibits consistent behavior across layers in both data-abundant and data-limited settings. In particular, the overall trends and the location of the semantic window both remain stable regardless of layer choice.

## C. Experimental Details

Unless stated otherwise, we apply mean pooling to obtain a single feature vector per sample. For EDM models, we average over spatial dimensions, mapping tensors of shape $N \times C \times H \times W$ to $N \times C$. For transformer-based models, we average over the token dimension, mapping $N \times T \times D$ to $N \times D$.

In all experiments in Section 5, we extract features at a fixed intermediate noise scale, using $\sigma_t = 0.29$ for EDM based models (Karras et al., 2022) and $t = 0.2$ for SiT based models (Ma et al., 2024).

**Figure 1.** We take a 2,000-image subset of ImageNet64 and extract representations from the `dec.16x16_block1` layer of a publicly available pretrained EDM model. For each image, this gives a tensor of shape $(576, 16, 16)$; we also extract representations for 14 augmented views (`hflip`, `shift±4x`, `shift±4y`, `cropc56`, `croptl5`, `cropbr54`, `bright±0.08`, `contrast0.85`, `sat0.85`, `cutout22`, `blur3`), for a total of 15 views. We flatten each tensor into a $147{,}456 = 576 \times 16 \times 16$-dimensional vector. For each image, we take the mean across the 15 view representations to obtain $s$ (shared by the image and its augmented views), and subtract $s$ from each view representation to obtain the nuisance component $\xi$ for that view.

Finally, we pick a reference image (could be augmented) and retrieve its nearest neighbors under cosine similarity, in terms of $s$ or $\xi$. We then plot the retrieved neighbors (could be augmented) in the first and second rows, respectively.

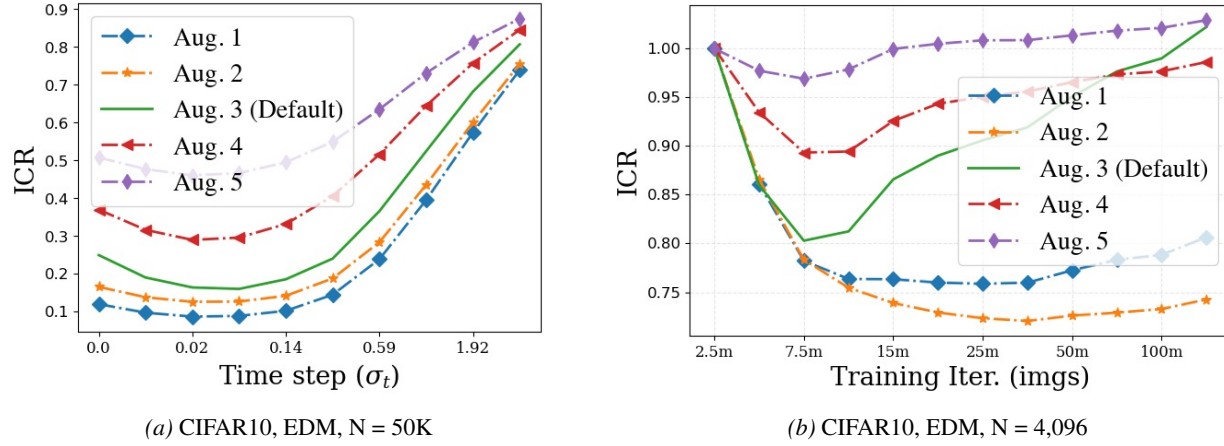

*(a)* CIFAR10, EDM, N = 50K          *(b)* CIFAR10, EDM, N = 4,096

*Figure 13.* **Sensitivity of** ICR **to augmentation design.** . We study the sensitivity of ICR to augmentation design by varying the strength of augmentations. We consider five augmentation levels and compute ICR under each setting. (a) In the data-abundant setting (50K training samples), the ICR curves across noise levels remain highly consistent across different augmentation strengths. (b) In the data-limited setting (4,096 training samples), the temporal evolution of ICR during training exhibits the same qualitative trend once sufficiently strong augmentations are applied. For clarity, we normalize each curve by its initial value to focus on the relative trend.

**Figure 2.** We evaluate ICR and linear probing accuracy using publicly available EDM models on CIFAR10 and a SiT-XL/2 model on ImageNet $256 \times 256$, together with a CIFAR100 EDM model that we train ourselves. For EDM backbones we extract features from the `dec.16x16_block1` layer near the bottleneck; for SiT-XL/2 we use the output of transformer block 14, the midpoint of the 28-layer network. For EDM models we train a logistic regression classifier with `scikit-learn` on the full set of training features and report accuracy on test features. For SiT-XL/2, due to the larger feature set, we subsample 200K training features (200 images per class) and train a linear classifier with AdamW (Loshchilov & Hutter, 2017) for 100 epochs (batch size 8192, learning rate $10^{-2}$, weight decay $10^{-4}$), reporting test accuracy at the final epoch. These hyperparameters are fixed across all noise levels and are not tuned, since our goal is to capture trends rather than optimize absolute performance. For computing ICR we use a random subset of $N = 4096$ training features for EDM models and $N = 20K$ for SiT-XL/2 at each noise level, with the subset held fixed across noise levels in each experiment.

**Figure 3.** We train an EDM model on CIFAR10 and a SiT-B/2 model on ImageNet $256 \times 256$ using the full training datasets and report FID together with ICR. FID is computed from 50K generated images for each experiment. As above, ICR is estimated from a subset of training features, using $N = 4096$ samples for the EDM model and $N = 20K$ for the SiT-B/2 model.

**Figure 4.** We train an EDM model on CIFAR10 using 4096 training images and report ICR together with the memorization ratio over the course of training. For the nearest neighbor visualizations on the right, we take generated samples and find their nearest neighbors among the training images directly in pixel space. For the snapshot at *Training iter = 200M*, we slightly cherry pick a few generated samples that are clearly memorized to highlight this effect; all other visualizations use fixed random seeds aligned with this snapshot to keep the plots consistent across training iterations.

**Figure 5.** We train an EDM model on ImageNet $64 \times 64$ using 10K training images (10 per class) and a SiT-B/2 model on ImageNet $256 \times 256$ using 20K training images (20 per class). Both models are trained with standard class conditional setups. When extracting features for ICR computation, to keep the procedure label-free, we encode all samples using the null class.

**Figure 6.** We follow the same pipeline as in Figure 1, but use checkpoints from an EDM model trained on only 10K ImageNet $64 \times 64$ samples. We visualize three typical phases of limited-data training: (i) early learning (first row), after 0.6M images, where the model is still improving; (ii) the onset of overfitting (second row), after 7M images, where ICR is smallest and the nearest neighbors remain meaningful and share structure with the reference; and (iii) severe overfitting (last row), after 50M images, where ICR increases and nearest neighbors are no longer semantic.

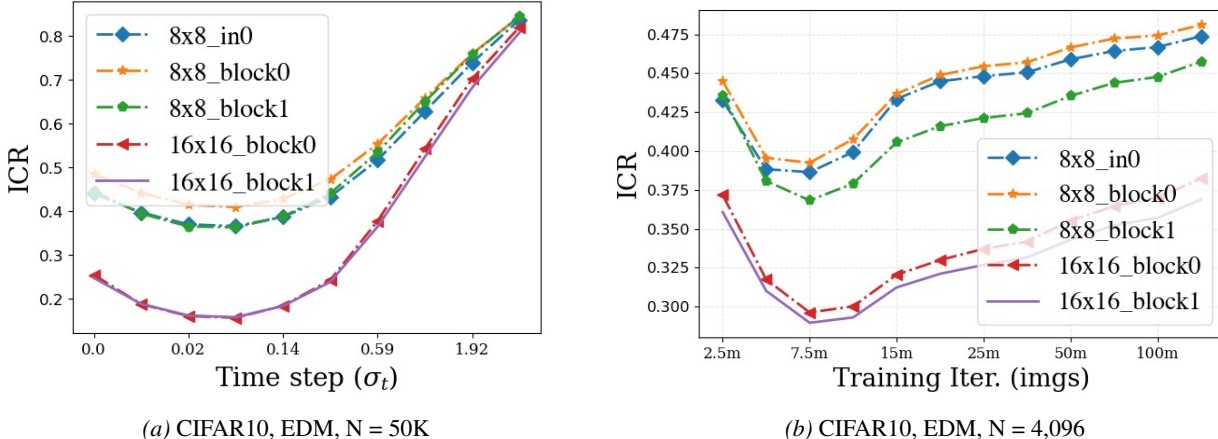

*(a)* CIFAR10, EDM, N = 50K    *(b)* CIFAR10, EDM, N = 4,096

*Figure 14.* **Sensitivity of ICR to the choice of feature extraction layer.** We select multiple layers around the middle of the diffusion model and compute ICR using features from each layer. (a) In the data-abundant setting (50K training samples), the trend of the ICR curves across noise levels remain highly consistent across different layer choices, with the location of the semantic window largely unchanged. (b) In the data-limited setting (4,096 training samples), the temporal evolution of ICR during training exhibits the same qualitative trend across layers, including the U-shaped behavior associated with memorization.

**Figure 7.** We reuse the EDM models trained on CIFAR10 with 4096 images and with the full 50K images to report the traces of $\boldsymbol{\Sigma}_s$ and $\boldsymbol{\Sigma}_\xi$ over training. To ensure that our notion of feature expansion is not conflated with simple growth in representation norms, we $\ell_2$ normalize each representation before computing the covariances.

## D. Alignment between Optimal Test Loss and ICR

In the main part of the paper, we discussed that ICR can be used as an early-stopping indicator when training diffusion models with limited data. In this section, we provide some preliminary theoretical insights for the underlying cause of such functionality. Specifically, through a simple Gaussian toy model, we show that ICR moves in the same direction as the Bayes optimal linear denoising loss.

**Toy two-layer linear model.** Fix a noise level $\sigma_t$ and an encoder $\boldsymbol{U} \in \mathbb{R}^{d \times D}$. Let $\boldsymbol{x}_t \in \mathbb{R}^D$ denote the noisy input and

$$\boldsymbol{h} \;=\; \boldsymbol{U}\boldsymbol{x}_t \in \mathbb{R}^d$$

be the feature. As in Section 3, we consider the invariance-variance decomposition of the feature $\boldsymbol{h} \;=\; \boldsymbol{s} + \boldsymbol{\xi}$ where $\boldsymbol{s}$ is the invariant component and $\boldsymbol{\xi}$ is the variant (residual) component induced by data augmentations and additive Gaussian noise. We denote their covariances by

$$\boldsymbol{\Sigma}_s(\boldsymbol{U}) \;=\; \mathrm{Cov}(\boldsymbol{s}), \qquad \boldsymbol{\Sigma}_\xi(\boldsymbol{U}) \;=\; \mathrm{Cov}(\boldsymbol{\xi}),$$

We reconstruct the clean image from the feature using a linear decoder $\boldsymbol{W} \in \mathbb{R}^{D \times d}$,

$$\widehat{\boldsymbol{x}}_0 \;=\; \boldsymbol{W}\boldsymbol{h},$$

and define the population linear denoising loss $\mathcal{L}_{\mathrm{lin}}(\boldsymbol{W}; \boldsymbol{U}) \;:=\; \mathbb{E}\big\|\boldsymbol{x}_0 - \boldsymbol{W}\boldsymbol{h}\big\|^2$. and thus

$$\mathcal{L}_{\mathrm{lin}}^\star(\boldsymbol{U}) \;:=\; \min_{\boldsymbol{W}} \mathcal{L}_{\mathrm{lin}}(\boldsymbol{W}; \boldsymbol{U}) \tag{8}$$

for the optimal linear denoising loss. For analytical clarity, we assume that $(\boldsymbol{x}_0, \boldsymbol{h})$ are jointly Gaussian.

**Proposition D.1** (Monotonicity of Bayes optimal loss and ICR). *Let $\boldsymbol{U}_1, \boldsymbol{U}_2$ be two encoders at the same noise level $\sigma_t$, and denote their invariance and variance covariances by*

$$\big(\boldsymbol{\Sigma}_s(\boldsymbol{U}_k), \boldsymbol{\Sigma}_\xi(\boldsymbol{U}_k)\big), \qquad k \in \{1, 2\}.$$

*Assume the Gaussian model above holds for each encoder.*

**(More variant energy hurts).** *If the invariant covariance is the same* $\Sigma_s(U_1) = \Sigma_s(U_2)$, *and the variant covariance of* $U_2$ *dominates that of* $U_1$ *in PSD order,*

$$\Sigma_\xi(U_1) \preceq \Sigma_\xi(U_2),$$

*then the optimal linear denoising loss and ICR both increase:*

$$\mathcal{L}_{\text{lin}}^\star(U_1) \leq \mathcal{L}_{\text{lin}}^\star(U_2), \quad \text{ICR}(U_1) \leq \text{ICR}(U_2). \tag{9}$$

**(More invariant energy helps).** *If instead the variant covariance is the same* $\Sigma_\xi(U_1) = \Sigma_\xi(U_2)$, *and the invariant covariance of* $U_2$ *dominates that of* $U_1$,

$$\Sigma_s(U_1) \preceq \Sigma_s(U_2),$$

*then the optimal linear denoising loss and ICR both decrease:*

$$\mathcal{L}_{\text{lin}}^\star(U_1) \geq \mathcal{L}_{\text{lin}}^\star(U_2), \quad \text{ICR}(U_1) \geq \text{ICR}(U_2). \tag{10}$$

*Proof.* We first derive the Bayes optimal loss in the original feature basis. Recall that for any measurable $g : \mathbb{R}^d \to \mathbb{R}^D$,

$$\mathcal{L}(g; \Sigma_s, \Sigma_\xi) = \mathbb{E} \|x_0 - g(h)\|^2.$$

Let $h = s + \xi$ with $s \sim \mathcal{N}(0, \Sigma_s)$ and $\xi \sim \mathcal{N}(0, \Sigma_\xi)$ independent, and $x_0 = As + C\xi$ as in the statement.

We now derive the covariance of $x_0$ and $h$:

$$\Sigma_{hh} := \text{Cov}(h) = \text{Cov}(s) + \text{Cov}(\xi) = \Sigma_s + \Sigma_\xi,$$
$$\Sigma_{xx} := \text{Cov}(x_0) = \text{Cov}(As + C\xi) = A\Sigma_s A^\top + C\Sigma_\xi C^\top,$$
$$\Sigma_{xh} := \text{Cov}(x_0, h) = \text{Cov}(As + C\xi, s + \xi) = A\Sigma_s + C\Sigma_\xi.$$

Now consider

$$x_0 - g(h) = \underbrace{(x_0 - \mathbb{E}[x_0 \mid h])}_{c_1} + \underbrace{(\mathbb{E}[x_0 \mid h] - g(h))}_{c_2}.$$

With this notation, we have

$$\mathbb{E} \|x_0 - g(h)\|^2 = \mathbb{E}\left[(c_1 + c_2)^\top (c_1 + c_2)\right]$$
$$= \mathbb{E} \|x_0 - \mathbb{E}[x_0 \mid h]\|^2 + \mathbb{E} \|\mathbb{E}[x_0 \mid h] - g(h)\|^2$$
$$+ 2 \mathbb{E}\left[(x_0 - \mathbb{E}[x_0 \mid h])^\top (\mathbb{E}[x_0 \mid h] - g(h))\right].$$

Then we can show the cross term vanishes:

$$\mathbb{E}[c_1^\top c_2] = \mathbb{E}\left[\mathbb{E}[c_1^\top c_2 \mid h]\right] = \mathbb{E}\left[\mathbb{E}\left[(x_0 - \mathbb{E}[x_0 \mid h])^\top c_2(h) \mid h\right]\right]$$
$$= \mathbb{E}\left[\mathbb{E}[x_0 - \mathbb{E}[x_0 \mid h] \mid h]^\top c_2(h)\right]$$
$$= \mathbb{E}\left[(\mathbb{E}[x_0 \mid h] - \mathbb{E}[x_0 \mid h])^\top c_2(h)\right] = 0,$$

where in the first equality we use the law of total expectation and in the third equality we use the fact that $c_2$ only depends on $h$ and is thus constant inside the conditional expectation. Hence

$$\mathbb{E} \|x_0 - g(h)\|^2 = \mathbb{E} \|x_0 - \mathbb{E}[x_0 \mid h]\|^2 + \mathbb{E} \|\mathbb{E}[x_0 \mid h] - g(h)\|^2.$$

Therefore $\mathbb{E} \|x_0 - g(h)\|^2$ is minimized when $g(h) = \mathbb{E}[x_0 \mid h]$ with minimum value $\mathbb{E} \|x_0 - \mathbb{E}[x_0 \mid h]\|^2$.

Now with $\boldsymbol{x}_0, \boldsymbol{h}$ jointly Gaussian, and $\boldsymbol{\Sigma}_{hh}$ invertible[7], we have

$$\mathbb{E}\left[\boldsymbol{x}_0 \mid \boldsymbol{h}\right] = \boldsymbol{\Sigma}_{xh}\boldsymbol{\Sigma}_{hh}^{-1}\boldsymbol{h}.$$

Write $\boldsymbol{W} = \boldsymbol{\Sigma}_{xh}\boldsymbol{\Sigma}_{hh}^{-1}$, then

$$
\begin{aligned}
\mathcal{L}^\star\left(\boldsymbol{\Sigma}_s, \boldsymbol{\Sigma}_\xi\right) &= \mathbb{E}\left\|\boldsymbol{x}_0 - \mathbb{E}[\boldsymbol{x}_0 \mid \boldsymbol{h}]\right\|^2 = \mathbb{E}\left\|\boldsymbol{x}_0 - \boldsymbol{W}\boldsymbol{h}\right\|^2 \\
&= \mathbb{E}\left[\left(\boldsymbol{x}_0 - \boldsymbol{W}\boldsymbol{h}\right)^\top \left(\boldsymbol{x}_0 - \boldsymbol{W}\boldsymbol{h}\right)\right] \\
&= \operatorname{Tr}\left(\boldsymbol{\Sigma}_{xx}\right) - \operatorname{Tr}\left(\boldsymbol{\Sigma}_{xh}\boldsymbol{\Sigma}_{hh}^{-1}\boldsymbol{\Sigma}_{hx}\right).
\end{aligned}
$$

Now what's left to show are the two co-monotonic facts. Consider a directional perturbation $\Delta\boldsymbol{\Sigma}_\xi \succeq \boldsymbol{0}$. Then

$$\delta\boldsymbol{\Sigma}_{xx} = \boldsymbol{C}\Delta\boldsymbol{\Sigma}_\xi\boldsymbol{C}^\top, \qquad \delta\boldsymbol{\Sigma}_{xh} = \boldsymbol{C}\Delta\boldsymbol{\Sigma}_\xi, \qquad \delta\boldsymbol{\Sigma}_{hh} = \Delta\boldsymbol{\Sigma}_\xi.$$

Using $\delta(\boldsymbol{\Sigma}_{hh}^{-1}) = -\boldsymbol{\Sigma}_{hh}^{-1}(\delta\boldsymbol{\Sigma}_{hh})\boldsymbol{\Sigma}_{hh}^{-1}$ and differentiating

$$\mathcal{L}^\star = \operatorname{Tr}(\boldsymbol{\Sigma}_{xx}) - \operatorname{Tr}(\boldsymbol{\Sigma}_{xh}\boldsymbol{\Sigma}_{hh}^{-1}\boldsymbol{\Sigma}_{hx}),$$

We can then calculate

$$
\begin{aligned}
\delta\mathcal{L}^\star &= \operatorname{Tr}\left(\delta\boldsymbol{\Sigma}_{xx}\right) - \operatorname{Tr}\left(\delta\boldsymbol{\Sigma}_{xh}\,\boldsymbol{\Sigma}_{hh}^{-1}\boldsymbol{\Sigma}_{hx}\right) - \operatorname{Tr}\left(\boldsymbol{\Sigma}_{xh}\,\delta\left(\boldsymbol{\Sigma}_{hh}^{-1}\right)\boldsymbol{\Sigma}_{hx}\right) - \operatorname{Tr}\left(\boldsymbol{\Sigma}_{xh}\,\boldsymbol{\Sigma}_{hh}^{-1}\,\delta\boldsymbol{\Sigma}_{hx}\right) \\
&= \operatorname{Tr}\left(\boldsymbol{C}\,\Delta\boldsymbol{\Sigma}_\xi\,\boldsymbol{C}^\top\right) - \operatorname{Tr}\left(\boldsymbol{C}\,\Delta\boldsymbol{\Sigma}_\xi\,\boldsymbol{\Sigma}_{hh}^{-1}\boldsymbol{\Sigma}_{hx}\right) \\
&\quad - \operatorname{Tr}\left(\boldsymbol{\Sigma}_{xh}\left(-\boldsymbol{\Sigma}_{hh}^{-1}\Delta\boldsymbol{\Sigma}_\xi\boldsymbol{\Sigma}_{hh}^{-1}\right)\boldsymbol{\Sigma}_{hx}\right) - \operatorname{Tr}\left(\boldsymbol{\Sigma}_{xh}\,\boldsymbol{\Sigma}_{hh}^{-1}\Delta\boldsymbol{\Sigma}_\xi\boldsymbol{C}^\top\right) \\
&= \operatorname{Tr}\left(\boldsymbol{C}\,\Delta\boldsymbol{\Sigma}_\xi\,\boldsymbol{C}^\top\right) - \operatorname{Tr}\left(\boldsymbol{C}\,\Delta\boldsymbol{\Sigma}_\xi\,\boldsymbol{\Sigma}_{hh}^{-1}\boldsymbol{\Sigma}_{hx}\right) \\
&\quad + \operatorname{Tr}\left(\boldsymbol{\Sigma}_{xh}\boldsymbol{\Sigma}_{hh}^{-1}\Delta\boldsymbol{\Sigma}_\xi\boldsymbol{\Sigma}_{hh}^{-1}\boldsymbol{\Sigma}_{hx}\right) - \operatorname{Tr}\left(\boldsymbol{\Sigma}_{xh}\,\boldsymbol{\Sigma}_{hh}^{-1}\Delta\boldsymbol{\Sigma}_\xi\boldsymbol{C}^\top\right).
\end{aligned}
$$

Using cyclicity of the trace to move $\Delta\boldsymbol{\Sigma}_\xi$ to the left in each term,

$$
\begin{aligned}
\delta\mathcal{L}^\star &= \operatorname{Tr}\left(\Delta\boldsymbol{\Sigma}_\xi\,\boldsymbol{C}^\top\boldsymbol{C}\right) - \operatorname{Tr}\left(\Delta\boldsymbol{\Sigma}_\xi\,\boldsymbol{\Sigma}_{hh}^{-1}\boldsymbol{\Sigma}_{hx}\boldsymbol{C}\right) \\
&\quad + \operatorname{Tr}\left(\Delta\boldsymbol{\Sigma}_\xi\,\boldsymbol{\Sigma}_{hh}^{-1}\boldsymbol{\Sigma}_{hx}\boldsymbol{\Sigma}_{xh}\boldsymbol{\Sigma}_{hh}^{-1}\right) - \operatorname{Tr}\left(\Delta\boldsymbol{\Sigma}_\xi\,\boldsymbol{C}^\top\boldsymbol{\Sigma}_{xh}\boldsymbol{\Sigma}_{hh}^{-1}\right) \\
&= \operatorname{Tr}\left(\Delta\boldsymbol{\Sigma}_\xi\left[\boldsymbol{C}^\top\boldsymbol{C} - \boldsymbol{\Sigma}_{hh}^{-1}\boldsymbol{\Sigma}_{hx}\boldsymbol{C} + \boldsymbol{\Sigma}_{hh}^{-1}\boldsymbol{\Sigma}_{hx}\boldsymbol{\Sigma}_{xh}\boldsymbol{\Sigma}_{hh}^{-1} - \boldsymbol{C}^\top\boldsymbol{\Sigma}_{xh}\boldsymbol{\Sigma}_{hh}^{-1}\right]\right).
\end{aligned}
$$

Now define

$$\boldsymbol{M} := \boldsymbol{C} - \boldsymbol{\Sigma}_{xh}\boldsymbol{\Sigma}_{hh}^{-1}.$$

Then

$$
\begin{aligned}
\boldsymbol{M}^\top\boldsymbol{M} &= \left(\boldsymbol{C} - \boldsymbol{\Sigma}_{xh}\boldsymbol{\Sigma}_{hh}^{-1}\right)^\top \left(\boldsymbol{C} - \boldsymbol{\Sigma}_{xh}\boldsymbol{\Sigma}_{hh}^{-1}\right) \\
&= \left(\boldsymbol{C}^\top - \boldsymbol{\Sigma}_{hh}^{-1}\boldsymbol{\Sigma}_{hx}\right)\left(\boldsymbol{C} - \boldsymbol{\Sigma}_{xh}\boldsymbol{\Sigma}_{hh}^{-1}\right) \\
&= \boldsymbol{C}^\top\boldsymbol{C} - \boldsymbol{C}^\top\boldsymbol{\Sigma}_{xh}\boldsymbol{\Sigma}_{hh}^{-1} - \boldsymbol{\Sigma}_{hh}^{-1}\boldsymbol{\Sigma}_{hx}\boldsymbol{C} + \boldsymbol{\Sigma}_{hh}^{-1}\boldsymbol{\Sigma}_{hx}\boldsymbol{\Sigma}_{xh}\boldsymbol{\Sigma}_{hh}^{-1},
\end{aligned}
$$

which matches the bracket above. Hence

$$\delta\mathcal{L}^\star\left(\boldsymbol{\Sigma}_s, \boldsymbol{\Sigma}_\xi\right) = \operatorname{Tr}\left(\Delta\boldsymbol{\Sigma}_\xi\,\boldsymbol{M}^\top\boldsymbol{M}\right).$$

Since $\Delta\boldsymbol{\Sigma}_\xi \succeq \boldsymbol{0}$ and $\boldsymbol{M}^\top\boldsymbol{M} \succeq \boldsymbol{0}$, the product inside the trace is positive semidefinite and hence

$$\delta\mathcal{L}^\star\left(\boldsymbol{\Sigma}_s, \boldsymbol{\Sigma}_\xi\right) = \operatorname{Tr}\left(\Delta\boldsymbol{\Sigma}_\xi\,\boldsymbol{M}^\top\boldsymbol{M}\right) \geq 0.$$

---

[7]In practice, we can enforce this by adding a small $\tau\boldsymbol{I}$ term to $\boldsymbol{\Sigma}_\xi$.

Therefore, consider $\boldsymbol{\Sigma}_{\xi,1}, \boldsymbol{\Sigma}_{\xi,2} \succeq \mathbf{0}$ with $\boldsymbol{\Sigma}_{\xi,1} \preceq \boldsymbol{\Sigma}_{\xi,2}$, and define

$$\boldsymbol{\Sigma}_\xi(t) := \boldsymbol{\Sigma}_{\xi,1} + t\left(\boldsymbol{\Sigma}_{\xi,2} - \boldsymbol{\Sigma}_{\xi,1}\right), \quad t \in [0,1],$$

and $\phi(t) = \mathcal{L}^\star\left(\boldsymbol{\Sigma}_s, \boldsymbol{\Sigma}_\xi(t)\right)$. Then

$$\phi'(t) = \mathrm{Tr}\left(\left(\boldsymbol{\Sigma}_{\xi,2} - \boldsymbol{\Sigma}_{\xi,1}\right) \boldsymbol{M}^\top \boldsymbol{M}\right) \geq 0,$$

i.e., $\phi(t)$ is non-decreasing in $t$ and therefore

$$\mathcal{L}^\star(\boldsymbol{\Sigma}_s, \boldsymbol{\Sigma}_{\xi,1}) \leq \mathcal{L}^\star(\boldsymbol{\Sigma}_s, \boldsymbol{\Sigma}_{\xi,2}).$$

Now we consider ICR. Fix $\boldsymbol{V}$ as in the statement and define

$$\widetilde{\boldsymbol{\Sigma}}_s := \boldsymbol{V}^\top \boldsymbol{\Sigma}_s \boldsymbol{V}, \qquad \widetilde{\boldsymbol{\Sigma}}_\xi := \boldsymbol{V}^\top \boldsymbol{\Sigma}_\xi \boldsymbol{V}.$$

By cyclicity of the trace,

$$\mathrm{ICR}(\boldsymbol{\Sigma}_s, \boldsymbol{\Sigma}_\xi) = \frac{\mathrm{Tr}(\boldsymbol{V}^\top \boldsymbol{\Sigma}_\xi \boldsymbol{V})}{\mathrm{Tr}(\boldsymbol{V}^\top (\boldsymbol{\Sigma}_s + \boldsymbol{\Sigma}_\xi) \boldsymbol{V})} = \frac{\mathrm{Tr}(\widetilde{\boldsymbol{\Sigma}}_\xi)}{\mathrm{Tr}(\widetilde{\boldsymbol{\Sigma}}_s + \widetilde{\boldsymbol{\Sigma}}_\xi)}.$$

Fix $\boldsymbol{\Sigma}_s$ and consider the same path $\boldsymbol{\Sigma}_\xi(t) = \boldsymbol{\Sigma}_{\xi,1} + t(\boldsymbol{\Sigma}_{\xi,2} - \boldsymbol{\Sigma}_{\xi,1})$, and set

$$\widetilde{\boldsymbol{\Sigma}}_\xi(t) := \boldsymbol{V}^\top \boldsymbol{\Sigma}_\xi(t) \boldsymbol{V} = \boldsymbol{V}^\top \boldsymbol{\Sigma}_{\xi,1} \boldsymbol{V} + t\, \boldsymbol{V}^\top (\boldsymbol{\Sigma}_{\xi,2} - \boldsymbol{\Sigma}_{\xi,1}) \boldsymbol{V},$$
$$\alpha(t) := \mathrm{Tr}(\widetilde{\boldsymbol{\Sigma}}_\xi(t)), \qquad \beta := \mathrm{Tr}(\widetilde{\boldsymbol{\Sigma}}_s) > 0.$$

Then

$$\mathrm{ICR}(t) := \mathrm{ICR}(\boldsymbol{\Sigma}_s, \boldsymbol{\Sigma}_\xi(t)) = \frac{\alpha(t)}{\alpha(t) + \beta}.$$

Differentiating,

$$\alpha'(t) = \mathrm{Tr}\left(\boldsymbol{V}^\top (\boldsymbol{\Sigma}_{\xi,2} - \boldsymbol{\Sigma}_{\xi,1}) \boldsymbol{V}\right) = \mathrm{Tr}\left((\boldsymbol{\Sigma}_{\xi,2} - \boldsymbol{\Sigma}_{\xi,1}) \boldsymbol{V} \boldsymbol{V}^\top\right) \geq 0,$$

because $\boldsymbol{\Sigma}_{\xi,2} - \boldsymbol{\Sigma}_{\xi,1} \succeq \mathbf{0}$ and $\boldsymbol{V} \boldsymbol{V}^\top \succ \mathbf{0}$ imply

$$\mathrm{Tr}\left((\boldsymbol{\Sigma}_{\xi,2} - \boldsymbol{\Sigma}_{\xi,1}) \boldsymbol{V} \boldsymbol{V}^\top\right) = \mathrm{Tr}\left((\boldsymbol{V} \boldsymbol{V}^\top)^{1/2}(\boldsymbol{\Sigma}_{\xi,2} - \boldsymbol{\Sigma}_{\xi,1})(\boldsymbol{V} \boldsymbol{V}^\top)^{1/2}\right) \geq 0.$$

Hence

$$\mathrm{ICR}'(t) = \frac{\beta\, \alpha'(t)}{(\alpha(t) + \beta)^2} \geq 0,$$

so $\mathrm{ICR}(t)$ is non-decreasing in $t$ and we obtain

$$\mathrm{ICR}(\boldsymbol{\Sigma}_s, \boldsymbol{\Sigma}_{\xi,1}) \leq \mathrm{ICR}(\boldsymbol{\Sigma}_s, \boldsymbol{\Sigma}_{\xi,2}).$$

which completes the proof. □

