# OpenReview forum: "Evaluating the Representation Space of Diffusion Models via Self-Supervised Principles"
_ICML.cc/2026/Conference — ICML 2026 regular_

### Official Review · Reviewer_2dkk · 2026-03-08

**Soundness:** 4
**Presentation:** 4
**Significance:** 4
**Originality:** 4
**Overall Recommendation:** 5
**Confidence:** 4

**Summary:**

This work introduces a metric named Invariant Contamination Ratio (**ICR**), which measures how residual, augmentation-sensitive energy contaminates invariant signal in feature space. It is found that (1) ICR can spot when memorization arises during the training dynamics of a diffusion model with small data load, (2) it correlates well withe behavior of the FID metric which is typically used to evaluate generative image models, and (3) most importantly, ICR can spot which diffusion noise level corresponds well with down-stream performance obtained from taking a pre-trained diffusion backbone and use it to train a classifier.

**Compliance With Llm Reviewing Policy:**

Affirmed.

**Final Justification:**

Good work. I am happy with my current scores and I think they are appropriate with the current work and its writings!

**Key Questions For Authors:**

Before the questions, I saw that you used a different font of ICR. Perhaps, it's better to just bold it.

1. Your description of **Representation expansion** is a bit confusing to me. I think you are saying that the embeddings need to be diverse among different classes of things (or images in this case), and depending on the amount of noise, the embeddings should be able to **spread out** more if there is too much noise or remain stable if there exists little noise. My suggestion is to include a little bit more description or rephrasing.

2. For Eq. (5), what do you mean by generalized eigenvalues? Are they not from Eq. (4)? Also, do you need to use all of the $d$-eigenvalues for Eq. (5)? What if they are sparse in values?

3. When you train your diffusion model, at some small training data sizes, the model can produce a peculiar phenomenon called **spurious states** (see this [paper](https://arxiv.org/abs/2505.21777)). These patterns are stable attractors of the energy function, which the diffusion model learns. Sometimes, they appear visually as a mixture of training data points or prototypes, like some of your generated images in Fig. (4). Do you think your metric in some ways help identify the regime (with respect to noise-levels and training iterations) where prototypes occur (which is very useful for classification)?

**Limitations:**

See weakness (2). But overall, it's a good work.

**Strengths And Weaknesses:**

## Strengths

1. Good and interesting work. Specifically, the results are very interesting and fantastic. ICR, seems to me, could be a replacement or alternative to the FID metric. Also, one can utilize this metric and find which noise level or time $t$ corresponds well to good SSL *diffusion embeddings*.

2. The structure of the paper is good and the writing is clear. There are some minor issues with the writings, but overall it's easy to follow the paper and understand it.

## Weaknesses

1. There are a lot of works done on memorization and generalization in diffusion models and dynamical regimes of diffusion models, which you should cite. Especially, [[1]](https://arxiv.org/abs/2402.18491) and [[2]](https://arxiv.org/html/2505.16959v1).

2. The analyses are focused only on the EDM-based models. Although this might sound *pedantic* and *stupid*, it would be nice to see if the trend holds for other kinds of model, like non-score-based model like DDPM.

---

> ### Author Rebuttal · Authors · 2026-03-31
>
> We thank the reviewer for the positive and encouraging feedback. We are glad that the reviewer found the problem interesting, and appreciated the clarity of the presentation as well as the potential of ICR as a practical metric. We also thank the reviewer for the helpful suggestions and questions. Below, we address them in detail.
>
> > **On related work**.
>
> **Answer**: We thank the reviewer for pointing us to these valuable works. [1] analyzes the reverse diffusion process and identifies dynamical phases, including a collapse regime associated with memorization. Our work provides a complementary representation-level perspective, showing that this transition is reflected in the geometry of learned features and can be detected via ICR during training.
>
> [2] shows that diffusion models first generalize and then memorize, with memorization time scaling with dataset size. Our findings are consistent with this temporal view and further show that this transition can be identified from internal representations via ICR. We will expand the related work section accordingly.
>
> We will follow the reviewer’s suggestion to expand the related work section to include a more detailed discussion of these works and related literature on memorization and generalization.
>
> > **Model generality.**
>
> **Answer**: We thank the reviewer for the suggestion. We understand that DDPM is also a score-based diffusion model under an equivalent formulation, and our analysis is not restricted to a specific parameterization. In addition to EDM, we include transformer-based diffusion models (SiT), which adopt a different architecture and show consistent trends (Figure 5).
>
> We agree that evaluating additional diffusion variants such as DDPM would further strengthen the empirical evidence, and we will include such results in the revision.
>
> > **Question regarding representation expansion**.
>
> **Answer**: We thank the reviewer for the helpful suggestion and agree that this part can be clarified.
>
> By representation expansion, we do not refer to class-level diversity, but rather to how features spread in the representation space across data samples. More specifically, it describes the extent to which the feature covariance occupies multiple directions, i.e., whether the representation is low-rank or well spread in the embedding space. In our setting, It is characterized by the covariance $\Sigma_h = \Sigma_s + \Sigma_\xi$. As shown in Appendix B.1, $Tr(\Sigma_\xi)$ increases with noise, while $Tr(\Sigma_s)$ follows a unimodal trend. This indicates that representation quality is governed by the balance between invariant and residual components, rather than expansion alone.
>
> We will revise the manuscript to clarify this definition and improve the presentation.
>
> > **Generalized eigenvalues (Eq. 5).**
>
> **Answer**: We thank the reviewer for the careful question. Yes, the generalized eigenvalues used in Eq. (5) are exactly those from Eq. (4): they are defined by the generalized eigenproblem
> $\Sigma_s v_i = \lambda_i \Sigma_xi v_i$,
> and Eq. (4) gives the equivalent Rayleigh quotient characterization.
>
> That is, the generalized eigenvalues are precisely the optimal invariant-to-residual signal-to-noise ratios along the Fisher directions. We will revise the paper to make this equivalence more explicit.
>
> We use all eigenvalues in Eq. (5) because our goal is to summarize the overall invariant-to-residual balance across the full representation space, not only the top few directions. We use all eigenvalues to summarize the global representation quality. If the spectrum is sparse, it indicates that only a few directions carry strong invariant signal, while others are dominated by residual variation.
>
> > **Paper on spurious states / prototypes.**
>
> **Answer**: Thank you for this insightful question. The cited paper mainly study the near-convergence regime, whereas we focus more on the training dynamics. As training proceeds, we observe a similar progression: diffusion models move from generating lower-quality but novel samples to spurious, prototype-like samples, and finally to memorized samples. Our analysis focuses on how the learned representation structure drives the emergence of spurious samples and then memorization.
>
> Therefore, ICR may indeed help identify the regime where prototypes appear. We think prototypes emerge when the model is still trying to generate samples that are both novel and faithful, but the underlying information from the training set has been largely exhausted. At this stage, the model reaches its peak representation quality and tends to generate prototypes, which are also representative of the training set.
>
> We thank the reviewer again for the paper referenced, which suggests an interesting connection between good ICR, good classification accuracy, and the emergence of prototypes.
>
> [1] Biroli et al; Dynamical Regimes of Diffusion Models.
>
> [2] Favero and Sclocchi; Bigger Isn’t Always Memorizing: Early Stopping Overparameterized Diffusion Models.

---

> > ### Author Rebuttal · Reviewer_2dkk · 2026-04-01
> >
> > I am pretty happy with their paper. My main concerns were primarily some of the writings. I believe the authors' rebuttal clarifies my misunderstandings.

---

> > > ### Author Response · Authors · 2026-04-06
> > >
> > > Thank you very much for your thoughtful review and kind follow-up. We are glad the rebuttal helped clarify the concerns, and we really appreciate your time and consideration.

---

### Official Review · Reviewer_F5WS · 2026-03-12

**Soundness:** 3
**Presentation:** 3
**Significance:** 3
**Originality:** 3
**Overall Recommendation:** 5
**Confidence:** 4

**Summary:**

This work introduces Invariant Contamination Ratio (ICR), which is a label-free metric for evaluating the representation space of diffusion models using principles derived from self-supervised learning (SSL) literature. The metric is computed using two quantities: the invariant component ($s$, which captures the unchanged part from corruption) and the residual component ($\xi$, which captures variations from noisy views).


The paper shows that ICR can be used in:
 - Across the noise interval, it accurately predicts the noise level where the representation space of the diffusion model yields the highest accuracy for downstream classification tasks.
 - Across the training process and in the data limited settings, it highlights the transition from generalization to memorization in the training dynamic.

**Compliance With Llm Reviewing Policy:**

Affirmed.

**Final Justification:**

The authors addressed my concerns in the rebuttal. I think this is overall good work and I keep my positive assessment.

**Key Questions For Authors:**

1. Do the authors have results on how sensitive ICR is to the choice of the specific layers and the depth of the U-Net? In addition, do these changes shift the optimal "semantic window"?

2. How sensitive is the ICR metric to the specific suite of spatial augmentations chosen? Have the authors performed other ablation study on different datasets and different sets of perturbations?

3. Can the transition from invariant feature expansion to residual feature expansion is directly linked to the transition to memorization of specific training samples be formally shown?

**Limitations:**

Yes

**Strengths And Weaknesses:**

Strength:
 - Applying SSL principles of representation invariance and expansion to analyze diffusion models provides a novel perspective on the understanding of internal features learnt using denoising objectives.
 - The proposed metric ICR can be calculated solely with training samples and without labels or any other expensive computational steps.
 - The strong empirical evidence highlights that ICR successfully correlates with key outcomes. It consistently predicts the optimal noise level for linear probing and tracks the training dynamic of the diffusion model.

Weakness:
 - ICR relies on the heuristic of selecting the representation from a specific layer from the diffusion model. A thorough analysis of how the two components in ICR can be affected by the depth of the model would help better understand the interaction.
 - Although the experiments strongly suggest that ICR aligns with the memorization ratio in the data limited setting, the theoretical mechanism connecting residual components to the memorization effect is largely intuitive.
 - The experiments focus on CIFAR-10 and a two-view approximation for estimation of ICR, it is unclear how accurate is ICR on more complicated datasets.

---

> ### Author Rebuttal · Authors · 2026-03-31
>
> We thank the reviewer for the thoughtful and positive feedback. We are glad that the reviewer appreciated the use of self-supervised learning principles to analyze diffusion representations, as well as the simplicity and label-free nature of ICR. We also appreciate the recognition of the strong empirical results. Below, we address the reviewer’s questions and suggestions in detail.
>
> > Although the experiments strongly suggest that ICR aligns with the memorization ratio in the data limited setting, the theoretical mechanism connecting residual components to the memorization effect is largely intuitive. Can the transition from invariant feature expansion to residual feature expansion is directly linked to the transition to memorization of specific training samples be formally shown?
>
> **Answer**: We thank the reviewer for the insightful comment and acknowledge this limitation. We note that in Appendix D, we provide an initial theoretical analysis using a two-layer linear model, where we show that the optimal test denoising loss and ICR share the same monotonicity with respect to the residual component. This offers a first step toward formally linking residual-dominated representations to poorer generalization behavior.
>
> However, we feel that this analysis is quite limited due to the simplicity of the model, which is why we place it in the Appendix. And we believe that extending this connection to more realistic nonlinear settings is an important direction for future work.
>
> At the sample level, recent work [1] shows that memorized samples exhibit distinctive, less consistent feature representations compared to novel samples.  We conjecture that this observation is consistent with residual components becoming dominant during prolonged training in data-limited regimes, which aligns with the emergence of memorization.
>
> > Do the authors have results on how sensitive ICR is to the choice of the specific layers and the depth of the U-Net? In addition, do these changes shift the optimal "semantic window"? ICR relies on the heuristic of selecting the representation from a specific layer from the diffusion model. A thorough analysis of how the two components in ICR can be affected by the depth of the model would help better understand the interaction.
>
> **Answer**: We thank the reviewer for the important question. Following the reviewer’s suggestion, we conduct an ablation study to evaluate the sensitivity of ICR to layer selection. Specifically, we extract features from multiple layers around the middle of the diffusion model and compute ICR for each choice. The results (see [Link](https://anonymous.4open.science/r/ICML2026_Rebuttal-E476/LayerChoice.png)) show that ICR exhibits consistent behavior across layers in both data-abundant and data-limited settings. In particular, the overall trends and the location of the semantic window both remain stable regardless of layer choice.
>
> > The experiments focus on CIFAR-10 and a two-view approximation for estimation of ICR, it is unclear how accurate is ICR on more complicated datasets.
>
> **Answer**: We thank the reviewer for the helpful comment. We note that our experiments are not limited to CIFAR-10; the manuscript also includes results on ImageNet (Figures 3 and 5), which demonstrate consistent behavior of ICR on a more complex dataset.
>
> Following the reviewer’s suggestion, we further evaluate ICR on the LSUN Church dataset, which is more complex and unlabeled. The results (see [Link](https://anonymous.4open.science/r/ICML2026_Rebuttal-E476/LsunChurch2048.png)) show that the relationship between ICR and memorization remains consistent, indicating the label-free nature and generality of ICR. We will include the new experiment in the revision.
>
> > How sensitive is the ICR metric to the specific suite of spatial augmentations chosen? Have the authors performed other ablation study on different datasets and different sets of perturbations?
>
> **Answer**: We thank the reviewer for the insightful question. We study the sensitivity of ICR to perturbations by varying the strength of augmentations (see [Link](https://anonymous.4open.science/r/ICML2026_Rebuttal-E476/AugmentChoice.png)).
>
> In the data-abundant setting, ICR remains highly consistent across augmentation choices, with similar trends across noise levels. In the data-limited setting, we observe that once a sufficiently rich augmentation pipeline is used (e.g., random crop + flip + color jitter and related variants), the resulting ICR trends are also consistent.
>
> In contrast, when using overly weak augmentations (e.g., random crop only), the behavior becomes less stable. This is expected, as the invariant–residual decomposition relies on sufficiently diverse perturbations to meaningfully separate invariant and residual components.
>
> We will include this analysis and discussion in the revision.
>
> [1] Zhang et al; Generalization of Diffusion Models Arises with a Balanced Representation Space.

---

> > ### Author Rebuttal · Reviewer_F5WS · 2026-04-02
> >
> > Thanks to the authors for clarifying.

---

> > > ### Author Response · Authors · 2026-04-06
> > >
> > > Thank you for your thoughtful review and follow-up. We are glad that our rebuttal helped clarify the concerns, and we sincerely appreciate your time and feedback.

---

### Official Review · Reviewer_4DZQ · 2026-03-13

**Soundness:** 2
**Presentation:** 2
**Significance:** 2
**Originality:** 2
**Overall Recommendation:** 3
**Confidence:** 4

**Summary:**

This paper studies the representation space of diffusion models from a self-supervised learning perspective. The authors decompose diffusion features into an invariant component and a residual component induced by noise and augmentations, and propose the Invariant Contamination Ratio (ICR) to quantify their interaction. Using this framework, the paper analyzes representation invariance across noise levels and training dynamics, and investigates how representation statistics relate to downstream performance and memorization behavior during training.

**Compliance With Llm Reviewing Policy:**

Affirmed.

**Final Justification:**

The authors’ response addresses some of my clarification questions. However, my main concerns about the practical relevance and overall usefulness of the results remain.
	1.	I still do not see how this work, or the intuition behind it, would concretely guide future research, especially in data-rich settings. In particular, I would like to see at least one application of the proposed ICR or SNR framework to a practical problem, such as inference acceleration. For example, in that setting, it is not clear to me whether ICR is even the most appropriate notion.
	2.	There also remains a lack of theoretical justification, which makes me hesitant to raise my final score.

Therefore, I will keep my score unchanged.

**Key Questions For Authors:**

1. More details about the methodology and hyperparameter settings are needed for clarity and reproducibility (see Weakness 1).

2. The paper evaluates the proposed analysis through classification performance, generation quality, and memorization experiments. However, the implications of these results are not clearly articulated. It would be helpful if the authors could further discuss how these findings improve our understanding of generative models and what practical insights they provide for model design or training.

**Limitations:**

yes

**Strengths And Weaknesses:**

**Strength**
1. The proposed ICR metric can detect the transition from generalization to memorization by tracking residual energy along Fisher directions, providing a practical way to monitor memorization behavior from training representations without relying on external evaluators or held-out data.

2. The paper introduces a clear decomposition of diffusion features inspired by self-supervised learning, offering a structured way to study invariance and representation geometry across noise levels


**Weakness**
1. The description of the notation and methodology could be clearer. For example, in Section 3.2, it is unclear whether $h$ refers to a specific layer of the diffusion model or to a separate neural network. It is also not specified how the expectation $E_a$ is computed in practice. In addition, the meaning of $a(x)$ is ambiguous. Whether it corresponds to $x_t$ or to another degraded version of the input. Finally, it is unclear how $\sigma_t$ in Figure 2 is introduced into the model. Overall, an additional section describing the algorithmic procedure or implementation details would help improve clarity.

2. Figure 2 shows a correlation between ICR and classification accuracy, but the practical insight of this observation is not fully explained. In classification tasks, performance can already be directly evaluated using a validation or test set, so it is unclear what additional benefit the ICR-based analysis provides. Moreover, the optimal noise level $\sigma^*$ identified for classification tasks may not necessarily translate to generative tasks, since the objectives of the two tasks differ. This limitation should be discussed more carefully.

3. Figure 5(a) does not appear to fully support the paper’s claim, as the increase in ICR does not clearly align with the increase in the memory ratio. In the current Figures 4–5, two plots support the conclusion while one does not, which weakens the overall empirical evidence. Additional experiments on more datasets and model architectures would help strengthen the claim.

---

> ### Author Rebuttal · Authors · 2026-03-31
>
> We thank the reviewer for the detailed and constructive feedback. We are glad that the reviewer recognized the invariant–residual decomposition and the usefulness of ICR as a practical metric for detecting the transition from generalization to memorization. Due to space limitations, we summarize the reviewer’s comments and respond to each in detail below.
>
> >**Methodology clarity**.
>
> **Answer**:
> We thank the reviewer for the important question and clarify the key components:
>
> **Feature extractor** $h$: representations are extracted from a fixed intermediate layer of the diffusion backbone (no additional network). For U-Net, we use a bottleneck-adjacent layer with strong representation quality; for transformers, the middle block. Full experimental details are provided in Appendix C.
>
> **Expectation approximation**: we approximate the conditional expectation using a two-view scheme by sampling two independent perturbations per sample to estimate $\Sigma_s$ and $\Sigma_\xi$. We refer the reviewer to Appendix B.4 for more details.
>
> **Perturbation** $a(x_0)$:  a(x_0) denotes a stochastic perturbation consisting of (i) standard semantics-preserving data augmentations and (ii) additive Gaussian noise. Concretely, we first apply augmentations to $x_0$ and then add Gaussian noise to the augmented view to get $a(x_0)$. We agree that this was not sufficiently clear and will revise the manuscript accordingly.
>
> **Role of** $x_t$:  In diffusion models, inputs at different noise levels take the form $x_t = x_0 + \sigma_t * \epsilon$, where $\sigma_t$ controls the noise scale. We extract representations $h(x_t)$ at multiple noise levels, yielding a family of features with varying semantic quality. For each sigma_t, we compute ICR using training representations at that noise level. Figure 2 shows that ICR strongly correlates with classification performance across noise levels.
>
> We will follow the reviewer's advice to revise the manuscript to improve clarity and reproducibility.
>
> >**On Figure 2 and practical insight.**
>
> **Answer**:
> We agree that classification performance can be directly evaluated when labels are available. Our goal is to provide a complementary signal. ICR is computed solely from training representations, without labels, held-out data, or external evaluators, making it useful in settings where such signals are unavailable. Figure 2 serves two purposes: (1) validating that ICR correlates with downstream performance across noise levels, and (2) enabling its use as a proxy for representation quality in later analysis (training dynamics, memorization, and generation).
>
> >**On noise level selection.**
>
> **Answer:** We agree that the noise level optimal for classification may not coincide with that for generation. Our goal is not to claim a universal optimum, but to show that invariance exhibits a consistent unimodal structure across noise levels, inducing a range of noise levels where models learn strong invariant structures. This motivates fixing an intermediate noise level when studying generative behavior. We will clarify this limitation in the revision.
>
> >**On Figure 5 and memorization.**
>
> **Answer:**
> We clarify that our claim concerns consistent trends rather than pointwise alignment.
> (1) In data-limited regimes, ICR follows a U-shaped trajectory (decreasing then increasing), in contrast to the monotonic decrease in data-rich settings.
> (2) The onset of memorization consistently occurs after the minimum of ICR, indicating that ICR detects the transition earlier and can serve as a practical, easy-to-monitor early-stopping signal.
> Following the reviewer’s suggestion, we include additional results on the LSUN Church dataset (subset of 2048 images) (please refer to [Link](https://anonymous.4open.science/r/ICML2026_Rebuttal-E476/LsunChurch2048.png)), which show the same qualitative behavior.
>
> >**Implications.**
>
> **Answer**:
> We thank the reviewer for the insightful question.
> Conceptually, this work provides a representation-level perspective on generalization and memorization in diffusion models. Rather than relying solely on generated samples, we show that these behaviors are reflected in internal representations. In particular, the transition from generalization to memorization corresponds to a shift from invariant-dominated to residual-dominated representations, offering an intrinsic view of training dynamics and enabling early detection of memorization via ICR.
>
> Our results further suggest that self-supervised learning principles, such as invariance and representation expansion, naturally emerge in diffusion models despite their different objectives, pointing to a shared notion of “good” representations across discriminative and generative settings. This is supported by recent work such as REPA [1].
>
> We will revise the manuscript to better highlight these conceptual and practical implications.
>
> [1] Yu et al; Representation Alignment for Generation: Training Diffusion Transformers Is Easier Than You Think.

---

> > ### Author Rebuttal · Reviewer_4DZQ · 2026-04-04
> >
> > I think I understand the notation $a(\cdot)$ better now.
> >
> >
> >
> > I still have the following questions:
> >
> >
> > 1. Regarding classification accuracy, my understanding is that, at different noise levels, the model may learn structures at different levels of granularity and difficulty. If so, the minimum of ICR does not necessarily correspond to the best classification accuracy, because downstream tasks such as classification may rely on structures at different granularities. In addition, in Figure 2(c), the best test accuracy does not coincide with the minimum of ICR. So I still feel that classification accuracy may not be the most appropriate validation metric here, and that this phenomenon could reflect overfitting unless it is supported by more comprehensive studies.
> >
> >
> > 2. Since most diffusion models are trained in data-rich settings, where ICR decreases monotonically, what practical insight can we gain from ICR in those cases?
> >
> >
> > 3. A further question is why the measure is defined as ICR, which seems more like a noise-to-signal ratio, rather than a signal-to-noise ratio. How would a signal-to-noise ratio style measure, such as $\sum \lambda_i$ or even $\frac{1}{d}\sum \lambda_i$, perform instead?
> >
> >
> >
> > 4. If we define the measure in a signal-to-noise ratio (SNR) fashion, would it be possible to use it to replace the naive SNR in DPM-Solver to accelerate sampling? This might also help clarify the practical value of the measure.
> >
> >
> > 5. In order to estimate the covariance matrices, how many paired samples are you using? What is the dimensionality (i am assuming that you are estimating the covariance matrices in the latent space).

---

> > > ### Author Response · Authors · 2026-04-06
> > >
> > > We thank the reviewer for the follow-up questions and are very grateful for the reviewer's engagement with the paper. We address each point below.
> > >
> > > > **On ICR vs. classification accuracy.**
> > >
> > > We thank the reviewer for the careful observation and insightful question. We agree that different noise levels capture structures at different granularities, as noted in recent studies [1,2]. We also agree that classification accuracy may not be an ideal metric here, as it depends on task-specific granularity, and we use it primarily as a simple validation signal to verify that ICR reflects intrinsic representation structure.
> > >
> > > The reviewer also raises an important point regarding overfitting. Prior work [2] shows that the unimodal (inverted-U) trend in classification accuracy across noise levels (as observed in Figure 2) reflects effective generalization, while deviations from this pattern may indicate overfitting.
> > >
> > > To test ICR in the overfitting regime, we conduct an additional experiment in the data-limited setting with prolonged training (see [link](https://anonymous.4open.science/r/ICML2026_Rebuttal-E476/ICRNoiseDataLimited.png)). In this regime, the diffusion model overfits and memorizes: classification accuracy loses the unimodal trend and instead decreases monotonically as noise increases. In contrast, ICR increases monotonically and maintains a clear negative correlation with classification accuracy. This shows that ICR continues to capture representation quality even when classification behavior departs from the typical generalization pattern. Together with the results in the main paper, this suggests that ICR reflects an intrinsic property of the representation.
> > >
> > > > **Practical value in data-rich settings.**
> > >
> > > We thank the reviewer for this excellent question. While ICR decreases monotonically in data-rich regimes, this behavior remains informative:
> > >
> > > (1) **Representation dynamics across regimes.**
> > > Under sufficient data, representations progressively become more invariant-dominated throughout training, providing a clear view of representation evolution in a stable generalization regime. In contrast, in data-limited settings, ICR exhibits a U-shaped trajectory and signals the onset of memorization. Together, these observations suggest that ICR provides a unified, **representation-level perspective** for monitoring diffusion model training across regimes, and helps connect diffusion training with principles from representation learning, supporting recent approaches such as REPA [3].
> > >
> > > (2) **Reference-free monitoring of training.**
> > > Unlike traditional metrics such as FID [4], which require a reference dataset, ICR is computed directly from the model’s internal representations. This enables a reference-free way to monitor training progress, especially in settings where reference data may be unavailable.
> > >
> > > In the revision, we will make these points clearer.
> > >
> > > > **ICR and signal-to-noise ratio.**
> > >
> > > ICR is directly related to a signal-to-noise ratio. Specifically, it can be written as
> > > ICR = 1 / (1 + SNR), where SNR is defined as $\frac{1}{d} \sum \lambda_i.$
> > > Thus, using SNR or ICR is equivalent up to a monotonic transformation. We adopt the ICR form because it is bounded in (0,1).
> > >
> > > > **Relation to DPM-Solver.**
> > >
> > > This is an interesting direction. DPM-solver utilizes the concept of SNR to better implement efficient high-order solver for sampling, which is more like a auxilarly variable explicitly defined by the forward process, and model-agnostic. In our case, ICR reflects the learned rep. quality of model, but we hypothesize that it can also be used to guide and accelerate the scheduling and sampling, as in DPM-solver.
> > >
> > > > **Estimation details.**
> > >
> > > We thank the reviewer for the opportunity to clarify. Indeed we estimate the covariance matrices in the latent feature space by applying mean pooling:
> > >
> > > * EDM models: average over spatial dimensions, mapping $N \times D \times H \times W$ to $N \times D$.
> > > * Transformer-based models (SiT): average over the token dimension, mapping $N \times T \times D$ to $N \times D$.
> > >
> > > The resulting feature dimensions are D = 256 for EDM (CIFAR), D = 576 for EDM (ImageNet64), D = 768 for SiT-B/2, and D = 1152 for SiT-XL/2.
> > >
> > > For all experiments, $N$ equals the full training set size in data-limited settings and a large subset in data-abundant settings. Under standard concentration results for subgaussian features, the estimation error scales as $\mathcal{O}(\sqrt{D/N})$ in operator norm. We validate this in Appendix B.4 (Figure 10) by varying $N$, observing stable covariance estimates.
> > >
> > > [1] Diffusion Models Generate Images Like Painters: an Analytical Theory of Outline First, Details Later.
> > >
> > > [2] Understanding Representation Dynamics of Diffusion Models via Low-Dimensional Modeling.
> > >
> > > [3] Representation alignment for generation: Training diffusion transformers is easier than you think.
> > >
> > > [4] Exposing flaws of generative model evaluation metrics and their unfair treatment of diffusion models.

---

### Official Review · Reviewer_mF3E · 2026-03-13

**Soundness:** 3
**Presentation:** 4
**Significance:** 3
**Originality:** 3
**Overall Recommendation:** 3
**Confidence:** 3

**Summary:**

This paper studies the representation space of diffusion models from a self-supervised learning perspective. The authors propose a decomposition of diffusion representations into invariant and residual components and introduce a label-free metric, the Invariant Contamination Ratio (ICR), to measure how much perturbation-sensitive variation contaminates invariant signal in the feature space. Using this metric, the paper analyzes how representations evolve across diffusion noise levels and during training. The experiments show that ICR correlates with downstream classification performance across noise levels and can reveal an early learning phenomenon in limited-data settings.

**Compliance With Llm Reviewing Policy:**

Affirmed.

**Key Questions For Authors:**

The proposed metric involves several design choices, including the invariant-residual decomposition and the aggregation of generalized eigenvalues. Did the authors explore alternative formulations, and if so, how sensitive are the results to these choices?

**Limitations:**

yes

**Strengths And Weaknesses:**

## Strengths

- **Important and interesting problem.**

  The paper studies the representation space of diffusion models from a self-supervised learning perspective. Understanding why diffusion representations work well is an important and timely problem.

- **Simple yet effective metric.**

  The proposed ICR metric is simple and label-free, making it appealing in practice. If the observed behavior is robust, the metric could be useful for monitoring training.

- **Insightful empirical observations.**

  The experiments provide an intuitive picture of representation behavior across noise levels. The finding that intermediate noise levels yield more semantic representations aligns well with intuition and is insightful.

---

## Weaknesses
- **Limited theoretical justification of the metric.**

  While the metric is intuitive, the specific design choice is not fully justified. Other aggregation schemes might lead to similar behavior.

- **Limited experimental comparison with existing representation metrics.**

  The paper discusses Alignment/Uniformity but provides limited empirical comparison with other representation evaluation metrics.

- **Potential dependence on augmentation design.**

  Since the invariant–residual decomposition depends on the perturbation distribution, it would be helpful to analyze how sensitive the metric is to different augmentation choices.

---

> ### Author Rebuttal · Authors · 2026-03-31
>
> We thank the reviewer for the thoughtful and constructive feedback. We are glad that the reviewer found the problem important and the empirical results insightful. We also appreciate the comments on methodology justification, metric comparison, and augmentation sensitivity, and address them below.
>
> >Limited theoretical justification of the metric. While the metric is intuitive, the specific design choice is not fully justified. Other aggregation schemes might lead to similar behavior. The proposed metric involves several design choices, including the invariant-residual decomposition and the aggregation of generalized eigenvalues. Did the authors explore alternative formulations, and if so, how sensitive are the results to these choices?
>
> **Answer**: We thank the reviewer for the insightful question. We note that the design of ICR is guided by two key principles:
>
> (1) **Relative invariance**.
>
> We aim to measure the balance between invariant signal and residual variation, rather than absolute feature magnitude. This motivates a Fisher-style signal-to-noise ratio based on generalized eigenvalues, where each eigenvalue captures the invariant-to-residual ratio along one direction.
>
> A natural alternative is a trace-based ratio such as $Tr(\Sigma_\xi) / Tr(\Sigma_s)$, which aggregates all directions into a single global statistic. However, this aggregation loses directional information. For example, consider a representation where only a small number of directions carry strong invariant signal while the remaining directions are dominated by residual variation. In this case, the trace ratio averages over all directions and fails to reflect the presence of these highly informative directions. In contrast, the generalized eigenvalue formulation explicitly captures the signal-to-noise ratio along each direction and is therefore sensitive to such anisotropic structures.
>
> (2) **Stability under representation expansion.**
>
>  ICR depends on the relative covariance structure and is therefore robust to how representations spread during training. In contrast, distance-based metrics (e.g., alignment) depend on absolute distances and can behave inconsistently. As training progresses, representations expand to occupy more directions (Figure 7), which can cause distance-based metrics to increase even when representation quality improves (which we discussed with more detail in Appendix B.2).
>
> We will incorporate the reviewer’s suggestion to discuss the alternative formulation in the revision.
>
> >Limited experimental comparison with existing representation metrics. The paper discusses Alignment/Uniformity but provides limited empirical comparison with other representation evaluation metrics.
>
> **Answer**: We appreciate the reviewer for the important question. In addition to the discussion on Alignment/Uniformity (Appendix B.2), we further compare ICR with two representative metrics: (i) class separation [1], a supervised metric that relies on ground-truth labels, and (ii) Silhouette Score, which depends on pseudo-labels (e.g., from k-means) and is therefore sensitive to clustering choices.
>
> We redo the experiments in Figure 4 using these metrics and report the results in [Link](https://anonymous.4open.science/r/ICML2026_Rebuttal-E476/OtherMetrics.png). We observe that ICR aligns more consistently with the memorization ratio, while the other metrics fail to capture the same trend reliably.
>
> Importantly, ICR is fully label-free and does not require clustering or external supervision, making it applicable in settings where labels or reliable pseudo-labels are unavailable. We will include these additional comparisons and discussion in the revision.
>
> >Potential dependence on augmentation design.
> Since the invariant–residual decomposition depends on the perturbation distribution, it would be helpful to analyze how sensitive the metric is to different augmentation choices.
>
> **Answer**: We thank the reviewer for the insightful question. We study the sensitivity of ICR to augmentation design by varying the strength of augmentations (see [Link](https://anonymous.4open.science/r/ICML2026_Rebuttal-E476/AugmentChoice.png)).
>
> We observe that once a reasonably rich augmentation pipeline is used (e.g., random crop + flip + color jitter, and other variants), the resulting ICR trends are highly consistent, indicating that our findings are robust to the specific choice of augmentations.
> On the other hand, when using overly weak augmentations (e.g., random crop only), the behavior becomes less stable. This is expected, as the invariant–residual decomposition relies on sufficiently diverse perturbations to meaningfully separate invariant and residual components.
>
> We will follow the reviewer’s suggestion to include the discussion on augmentation choices in the revision.
>
> [1] Kornblith et al; Why Do Better Loss Functions Lead to Less Transferable Features?.

---

> > ### Author Rebuttal · Reviewer_mF3E · 2026-04-01
> >
> > Thank you for the thorough clarifications and the additional experiments, which I greatly appreciate. The new experimental results address several of my concerns and strengthen the empirical evaluation. However, the main issue regarding the limited theoretical justification remains unresolved. Therefore, I will maintain my current score.

---

> > > ### Author Response · Authors · 2026-04-06
> > >
> > > We sincerely thank the reviewer for recognizing the strengthened empirical evaluation provided in our rebuttal.
> > >
> > > Regarding the theoretical justification, we would like to clarify that we have made an initial step in Appendix D. In particular, using a two-layer linear model, we show that both the optimal test denoising loss and ICR depend monotonically on the residual component. This establishes a formal connection between residual-dominated representations and degraded generalization, consistent with our empirical observations.
> > >
> > > We agree that this analysis is limited to a simplified setting, and that a complete theoretical understanding for nonlinear diffusion models remains an open challenge. Nevertheless, we believe this result provides a meaningful first step, and we will clarify both the scope and limitations of our analysis in the revision.
> > >
> > > We thank the reviewer again for the constructive feedback and engaging discussion, which will help us further improve the paper.

---

### Decision · Program_Chairs · 2026-04-30

**Decision:**

Accept (regular)

**Comment:**

### Strengths

- The paper tackles an important question: why diffusion-model representations work well and how their internal geometry evolves across noise scales and training.
- The core perspective is novel and interesting: bringing self-supervised learning principles such as invariance and representation expansion into the analysis of diffusion features.
- The proposed ICR metric is simple, label-free, and potentially useful in practice.
- Reviewers found the observations about an intermediate semantic window and the use of ICR to track memorization dynamics insightful.

### Weaknesses

- The main concern is theoretical justification. The metric is intuitive and the rebuttal pointed to an initial analysis in Appendix D for a simplified linear setting.
- The practical implications are not yet fully clear. One reviewer commented that the paper falls short of showing how ICR would concretely guide future modeling or training in data-rich settings.

### Rebuttal and Remaining Concerns

The rebuttal helped positively. The authors clarified the notation and implementation details, added discussion of alternative metric formulations, and provided further empirical comparisons. These responses appear to have fully addressed the concerns of the two accepting reviewers and strengthened the paper’s overall empirical support.

At the same time, the rebuttal did not completely resolve the concerns of the two weak-reject reviewers. The central remaining issues are the limited theoretical justification for ICR and the still somewhat incomplete story about its practical value. While it is appealing to supply stronger theoretical justifications, it is acceptable to consider the simplified linear setting. Yet, it is encouraged to provide further discussions on the practicality of the ICR for training and steering diffusion models.